# TOPOLOGICAL EXPRESSIVITY OF RELU NEURAL NETWORKS

## ABSTRACT

We study the expressivity of ReLU neural networks in the setting of a binary classification problem from a topological perspective. Recently, empirical studies showed that neural networks operate by changing topology, transforming a topologically complicated data set into a topologically simpler one as it passes through the layers. This topological simplification has been measured by Betti numbers, which are algebraic invariants of a topological space. We use the same measure to establish lower and upper bounds on the topological simplification a ReLU neural network can achieve with a given architecture. We therefore contribute to a better understanding of the expressivity of ReLU neural networks in the context of binary classification problems by shedding light on their ability to capture the underlying topological structure of the data. In particular the results show that deep ReLU neural networks are exponentially more powerful than shallow ones in terms of topological simplification. This provides a mathematically rigorous explanation why deeper networks are better equipped to handle complex and topologically rich datasets.

## 1 INTRODUCTION

Neural networks are at the core of many AI applications. A crucial task when working with neural networks is selecting the appropriate architecture to effectively tackle a given problem. Therefore, it is of fundamental interest to understand the range of problems that can be solved by neural networks with a given architecture, i.e., its *expressivity*.

In recent years, many theoretical findings have shed light on the expressivity of neural networks. Universal approximation theorems (Cybenko, 1989) (Hornik, 1991) state that one hidden layer is already sufficient to approximate any continuous function with arbitrary accuracy. On the other hand, it is known that deep networks can represent more complex functions than their shallow counterparts, see e.g. (Telgarsky, 2016; Eldan and Shamir, 2016; Arora et al., 2018).

The measure of expressivity of a neural network should always be related to the problem it has to solve. A common scenario in which neural networks are employed is the binary classification problem, where the network serves as a classifier for a binary labeled dataset. Since topological data analysis has revealed that data often has nontrivial topology, it is important to consider the topological structure of the data when dealing with a binary classification problem. Naitzat et al. (2020) show through empirical methods that neural networks operate topologically, transforming a topologically complicated dataset into a topologically simple one as it passes through the layers. Given a binary labeled dataset, they assume that the positively labeled and the negatively labeled points are sampled from topological spaces $M_a$ and $M_b$ respectively that are entangled with each other in a nontrivial way. Their experiments show that a well-trained neural network gradually disentangles the topological spaces until they are linearly separable in the end, i.e, the space $M_b$ is mapped to the positive real line and $M_a$ to the negative real line. From a theoretical point of view, it is of interest to determine the extent of "topological change" that can be achieved by neural networks of a particular architecture. The topological expressivity of a neural network can therefore be measured by the complexity of the most complex topological spaces it can separate and is directly related to the complexity of the binary classification problem.

In this paper we investigate the topological expressivity of ReLU neural networks, which are one of the most commonly used types of neural networks (Glorot et al., 2011; Goodfellow et al., 2016). A

$(L + 1)$-*layer neural network (NN)* is defined by $L + 1$ affine transformations $T_\ell \colon \mathbb{R}^{n_{\ell-1}} \to \mathbb{R}^{n_\ell}$, $x \mapsto A_\ell x + b_\ell$ for $A_\ell \in \mathbb{R}^{n_{\ell-1} \times n_\ell}$, $b_\ell \in \mathbb{R}^{n_\ell}$ and $\ell = 1, \ldots, L+1$. The tuple $(n_0, n_1, \ldots, n_L, n_{L+1})$ is called the *architecture*, $L + 1$ the *depth*, $n_\ell$ the *width of the $\ell$-layer*, $\max\{n_1, \ldots, n_L\}$ the *width* of the NN and $\sum_{\ell=1}^{L} n_\ell$ the *size* of the NN. The entries of $A_\ell$ and $b_\ell$ for $\ell = 1, ..., L+1$ are called *weights* of the NN and the vector space of all possible weights is called the *parameter space* of an architecture. A ReLU neural network computes the function

$$F = T_{L+1} \circ \sigma_{n_L} \circ T_L \circ \cdots \circ \sigma_{n_1} \circ T_1,$$

where $\sigma_n \colon \mathbb{R}^n \to \mathbb{R}^n$ is the *ReLU function* given by $\sigma_n(x) = (\max(0, x_1), \ldots, \max(0, x_n))$.

Note that the function $F$ is piecewise linear and continuous. In fact, it is known that any continuous piecewise linear function $F$ can be computed by a ReLU neural network (Arora et al., 2018). However, for a fixed architecture $A$, the class $\mathcal{F}_A$ of piecewise linear functions that is representable by this architecture is not known (Hertrich et al., 2021; Haase et al., 2023). Conveniently, in the setting of a binary classification problem we are merely interested in the *decision regions*, i.e., $F^{-1}((-\infty, 0))$ and $F^{-1}((0, \infty))$ rather than the continuous piecewise linear function $F$ itself.

A common choice to measure the complexity of a topological space $X$ is the use of algebraic invariants. Homology groups are the essential algebraic structures with which topological data analysis analyzes data (Dey and Wang, 2022) and hence Betti numbers as the ranks of these groups are the obvious measure of topological expressivity. Intuitively, the $k$-th Betti number $\beta_k(X)$ corresponds to the number of $(k + 1)$-dimensional holes in the space $X$ for $k > 0$ and $\beta_0(X)$ corresponds to the number of path-connected components of $X$. Thus, one can argue that when a space (the support of one class of the data) has many connected components and higher dimensional holes, it is more difficult to separate this space from the rest of the ambient space, e.g., mapping it to the negative line. In Appendix 5.1.2 we present a brief introduction to homology groups. For an in-depth discussion of the aforementioned concepts, we refer to (Hatcher, 2002).

In order to properly separate $M_a$ and $M_b$, the sublevel set $F^{-1}((-\infty, 0))$ of the function $F$ computed by the neural network should have the same topological complexity as $M_a$. Bianchini and Scarselli (2014) measured the topological complexity of the decision region $F^{-1}((-\infty, 0))$ with the sum of all its Betti numbers. This notion of topological expressivity does not differentiate between connected components and higher dimensional holes. On the other hand, if an architecture is not capable of expressing the Betti numbers of different dimensions of the underlying topological space of the dataset, then for every $F \in \mathcal{F}_A$ there is a set of data points $U$ such that $F$ misclassifies every $x \in U$ (Guss and Salakhutdinov, 2018). Therefore it is of fundamental interest to understand each Betti number of the decision regions and hence we propose the following definition:

**Definition 1.** *The* topological expressivity *of a ReLU neural network* $F \colon \mathbb{R}^d \to \mathbb{R}$ *is defined as the vector* $\beta(F) = (\beta_k(F))_{k=0,\ldots,d-1} = (\beta_k(F^{-1}((-\infty, 0))))_{k=0,\ldots,d-1}$.

## 1.1 MAIN RESULTS

Our main contribution consists of lower and upper bounds on the topological expressivity for ReLU NNs with architectures. These bounds demonstrate that the growth of Betti numbers in neural networks depends on their depth. With an unbounded depth, Betti numbers in every dimension can exhibit exponential growth as the network size increases. However, in the case of a shallow neural network, where the depth remains constant, the Betti numbers of the sublevel set are polynomially bounded in size. This implies that increasing the width of a network while keeping the depth constant prevents exponential growth in the Betti numbers. Consequently, if a dataset possesses exponentially high Betti numbers (parameterized by some parameter $p$), accurate modeling of the dataset requires a deep neural network when the size of the neural network is constrained to be polynomial in parameter $p$ since the topological expressivity serves, as discussed above, as a bottleneck measure for effective data representation.

In Theorem 9, the lower bounds for the topological expressivity are given by an explicit formula, from which we can derive the following asymptotic lower bounds:

**Corollary 1.** *Let* $A = (d, n_1, \ldots, n_L, 1)$ *with* $n_L \geq 4d$ *and* $M = 2 \cdot \prod_{\ell=1}^{L-1} \lfloor \frac{n_\ell}{2d} \rfloor$, *then there is a ReLU NN* $F \colon \mathbb{R}^d \mapsto \mathbb{R}$ *with architecture* $A$ *such that*

    *(i)* $\beta_0(F) \in \Omega(M^d \cdot n_L)$

*(ii)* $\beta_k(F) \in \Omega(M^k \cdot n_L)$ *for* $0 < k < d$.

*In particular, given* $\boldsymbol{v} = (v_1, \ldots, v_d) \in \mathbb{N}^{d-1}$, *there is a ReLU NN $F$ of size* $O\left(\log\left(\sum_{k=1}^{d-1} v_k\right)\right)$ *such that* $\beta_k(F) \geq v_{k+1}$ *for all* $k \in \{0, \ldots, d-1\}$.

Corollary 1 provides a proof for a conjecture on lower bounds for the zeroth Betti number of the decision region given in (Guss and Salakhutdinov, 2018); in fact, it generalizes the statement to arbitrary dimensions. Furthermore, we observe that $L = 2$ hidden layers are already sufficient to increase the topological expressivity as much as we want at the expense of an increased width due to the above lower bound.

**Corollary 2.** *Given* $v \in \mathbb{N}^d$, *there exists an NN* $F \colon \mathbb{R}^d \to \mathbb{R}$ *of depth 2 such that* $\beta_k(F) \geq v_{k+1}$ *for all* $k \in \{0, \ldots, d-1\}$.

We obtain the lower bound by making choices for the weights of the NN, nevertheless, we show that our construction is robust with respect to small perturbations. In fact, in Proposition 10 we prove that we actually have an open set in the parameter space such that the respective functions all have the same topological expressivity.

Using an upper bound on the number of linear regions (Serra et al., 2017), we obtain the following upper bound on $\beta_k(F)$.

**Proposition 3.** *Let* $F \colon \mathbb{R}^d \to \mathbb{R}$ *be a neural network of architecture* $(d, n_1, \ldots, n_L, 1)$. *Then it holds that* $\beta_0(F) \leq \sum_{(j_1, \ldots, j_L) \in J} \prod_{\ell=1}^{L} \binom{n_\ell}{j_\ell}$ *and for all* $k \in [d-1]$ *that*

$$\beta_k(F) \leq \binom{\sum_{(j_1, \ldots, j_L) \in J} \prod_{\ell=1}^{L} \binom{n_\ell}{j_\ell}}{d - k},$$

*where* $J = \left\{(j_1, \ldots, j_L) \in \mathbb{Z}^L \colon 0 \leq j_\ell \leq \min\{d, n_1 - j_1, \ldots, n_{\ell-1} - j_{\ell-1}\} \text{ for all } \ell = 1, \ldots, L\right\}$.

If all hidden layers have dimension $n$, then for all $k \in [d-1]$ it holds that $\beta_k(F) \in O(n^{L \cdot d \cdot (d-k)})$ and $\beta_0(F^{-1}((-\infty, 0))) \in O(n^{L \cdot d})$ and hence is polynomially bounded in the width. By combining Corollary 1 and the latter fact, we can conclude that there is an exponential gap in the topological expressivity between shallow and deep neural networks. This aligns with other popular measures of expressivity, such as the number of linear regions, where similar exponential gaps are known (Serra et al., 2017; Montúfar et al., 2014; Montufar, 2017).

## 1.2 RELATED WORK

### 1.2.1 TOPOLOGY AND NEURAL NETWORKS

Recently, there is a vast stream of research studying neural networks by means of topology using empirical methods (Petri and Leitão, 2020; Guss and Salakhutdinov, 2018; Naitzat et al., 2020; Li et al., 2020) as well as from a theoretical perspective (Basri and Jacobs, 2017; Melodia and Lenz, 2020; Grigsby and Lindsey, 2022; Bianchini and Scarselli, 2014; Grigsby et al., 2022; Hajij and Istvan, 2020). Bianchini and Scarselli (2014) were the first that used Betti numbers as a complexity measure for decision regions of neural networks. Their work studies NNs with sigmoidal activation functions and shows that there is an exponential gap with respect to the sum of Betti numbers between deep neural networks and neural networks with one hidden layer. However, there are no insights about distinct Betti numbers. In Guss and Salakhutdinov (2018), the decision regions of ReLU neural networks ares studied with empirical methods and an exponential gap for the zeroth Betti number is conjectured. Our results prove the conjecture and extend the results of Bianchini and Scarselli (2014) for the ReLU case (see Section 3 and Appendix). Furthermore topological characteristics such as connectivity or boundedness of the decision regions are also investigated in (Fawzi et al., 2018; Grigsby and Lindsey, 2022; Grigsby et al., 2022; Nguyen et al., 2018).

### 1.2.2 EXPRESSIVITY OF (RELU) NEURAL NETWORKS

In addition to the universal approximation theorems (Cybenko, 1989; Hornik, 1991), there is a significant amount of research on the expressivity of neural networks, e.g., indicating that deep neural

networks can be exponentially smaller in size than shallow ones. For ReLU neural networks, the number of linear regions is often used as a measure of complexity for the continuous piecewise linear (CPWL) function computed by the network. It is well established that deep ReLU neural networks can compute CPWL functions with exponentially more linear regions than shallow ones, based on various results such as lower and upper bounds on the number of linear regions for a given architecture (Montufar, 2017; Serra et al., 2017; Montúfar et al., 2014; Arora et al., 2018). We partially use techniques from their works to establish our bounds on topological expressivity, which offers the advantage of being directly related to the complexity of binary classification problems.

### 1.3 NOTATION AND DEFINITIONS

A function $F\colon \mathbb{R}^d \to \mathbb{R}^d$ is continuous piecewise linear (CPWL) if there is a polyhedral complex covering $\mathbb{R}^d$, such that $F$ is affine linear over each polyhedron of this complex. A linear region of $f$ is a maximal connected convex subspace $R$ such that $f$ is affine linear on $R$, i.e., a full-dimensional polyhedron of the complex.[1] For a survey on polyhedral theory in deep learning see Huchette et al. (2023), and for a general introduction to polyhedra we refer to Schrijver (1986).

We denote by $[n]$ the set $\{1, \ldots, n\}$ and by $[n]_0$ the set $\{0, \ldots, n\}$. We denote by $\pi_j\colon \mathbb{R}^d \to \mathbb{R}$ the projection onto the $j$-th component of $\mathbb{R}^d$ and by $p_j\colon \mathbb{R}^d \to \mathbb{R}^j$ the projection onto the first $j$ components.

A crucial part of our construction is decomposing a unit cube into a varying number of small cubes. Thereby, given $\mathbf{m} = (m_1, \ldots, m_L) \in \mathbb{N}^L$ and $M = \left(\prod_{\ell=1}^{L} m_\ell\right)$, the set $W_{i_1, \ldots, i_d}^{(L, \mathbf{m}, d)}$ is defined as the cube of volume $\frac{1}{M^d}$ with "upper right point" $\frac{1}{M} \cdot (i_1, \ldots, i_d)$, i.e., the cube $\prod_{k=1}^{d} [\frac{(i_k - 1)}{M}, \frac{i_k}{M}] \subset [0,1]^d$. The indices $(L, \mathbf{m}, d)$ are omitted whenever they are clear from the context.

We denote by $D^k = \{x \in \mathbb{R}^k\colon \|x\| < 1\}$ the $k$-dimensional standard open disk and by $S^k = \{x \in \mathbb{R}^{k+1}\colon \|x\| = 1\}$ the $k$-dimensional standard sphere. We consider these sets as "independent" topological spaces. Therefore, it is justified to abstain from picking a specific norm, since all norms on $\mathbb{R}^k$ are equivalent.

For $k, m \in \mathbb{N}$ with $m \leq k$, the *(j-dimensional open) k-annulus* is the product space $S^k \times D^{j-k}$. Note that since $S^k$ has one connected component and a $(k+1)$-dimensional hole, it holds that $\beta_0(S^k) = \beta_k(S^k) = 1$ and the remaining Betti numbers equal zero. The $j$-dimensional open $k$-annulus is an $j$-dimensional manifold that can be thought as a thickened $k$-sphere and hence its Betti numbers coincide with the ones from the $k$-sphere. In Appendix 5.1.2 the reader can find a more formal treatment of the latter fact.

In contrast to $D^k$ and $S^k$, which are only seen as spaces equipped with a topology, we also consider neighborhoods around certain points $x \in \mathbb{R}^d$ as subsets of $\mathbb{R}^d$. To make a clear distinction, we define the space $B_r^d(x)$ as the *d-dimensional open r-ball around $x$ with respect to the 1-norm*, i.e., the space $\{y \in \mathbb{R}^d\colon \|x - y\|_1 < r\}$. Note that for $r < r'$, the set $B_r^k(x) \setminus \overline{B_{r'}^k(x)}$ is homeomorphic to a $k$-dimensional open $(k-1)$-annulus and we will refer to them as $(k-1)$-annuli as well. These annuli will be the building blocks of our construction for the lower bound.

The rest of the paper is devoted to proving the lower and upper bounds. Most of the statements come with an explanation or an illustration. In addition, formal proofs for these statements are also provided in the appendix.

## 2 LOWER BOUND

In this section, our aim is to construct a neural network $F\colon \mathbb{R}^d \to \mathbb{R}$ of depth $L+2$ such that $\beta_k(F)$ grows exponentially in the size of the neural network for all $k \in [d-1]_0$.

We propose a construction that is restricted to architectures where the widths $n_1, \ldots, n_{L+1}$ of all hidden layers but the last one are divisible by $2d$. This construction, however, is generalized for any

---

[1]In the literature there exists also a slightly different definition of a linear region leaving out the necessity of the region being convex, but the bounds we use are all applicable to this definition of a linear region.

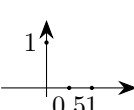

Figure 1: The graph of the function $\pi_j \circ h^{(1,2,d)}$ that folds the unit interval, i.e., mapping the interval $[0, 0.5]$ and $[0.5, 1]$ to the unit interval. This function is realised by a hidden layer with 2 hidden neurons.

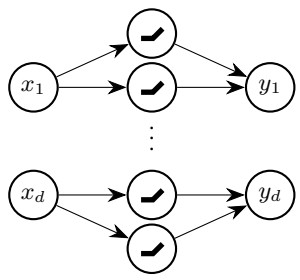

Figure 2: The architecture of the one hidden layer neural network $h^{(1,2,d)}$ that folds the $d$-dimensional unit cube by folding every component of the cube as described in Figure 1

architecture where the dimension of all hidden layers is at least $2d$ by inserting at most $2d$ auxiliary neurons at each layer at which a zero map is computed. Correspondingly, one obtains a lower bound by rounding down the width $n_\ell$ at each layer to the largest possible multiple of $2d$. In particular, a reduction to the case in Theorem 24 does not have an effect on the asymptotic size of the NN.

The key idea is to construct $F = f \circ h$ as a consecutive execution of two neural networks $f$ and $h$, where the map $h \colon \mathbb{R}^d \to \mathbb{R}^d$ is an ReLU NN with $L$ hidden layers that identifies exponentially many regions with each other. More precisely, $h$ cuts the unit cube of $\mathbb{R}^d$ into exponentially many small cubes $W_{i_1,\ldots,i_d} \in [0,1]^d$ and maps each of these cubes to the whole unit cube by scaling and mirroring. The one hidden layer ReLU NN $f$ then cuts the unit cube into pieces so that $f$ on the pieces alternatingly takes exclusively positive respectively negative values. Since $h$ maps all $W_{i_1,\ldots,i_d}$ to $[0,1]^d$ by scaling and mirroring, every $W_{i_1,\ldots,i_d}$ is cut into positive-valued and negative-valued regions by the composition $f \circ h$ in the same way as $[0,1]^d$ is mapped by $f$, up to mirroring. The cutting of the unit cube and the mirroring of the small cubes in the map to $[0,1]^d$ are chosen in such a way that the subspaces on which $F$ takes negative values form $k$-annuli for every $k \in [d-1]$. Since $h$ cuts the unit cube into exponentially many small cubes, we obtain exponentially many $k$-annuli for every $k \in [d-1]$ in the sub level set $F^{-1}((-\infty, 0))$.

The idea of constructing a ReLU neural network that folds the input space goes back to Montúfar et al. (2014), where the construction was used to show that a deep neural network with ReLU activation function can have exponentially many linear regions. Using their techniques, we first build a 1-hidden layer NN $h^{(1,m,d)} \colon \mathbb{R}^d \to \mathbb{R}^d$ for $m \in \mathbb{N}$ even that folds the input space, mapping $m^d$ many small cubes $W_{i_1,\ldots,i_d}^{(1,m,d)} \subset [0,1]^d$ by scaling and mirroring to $[0,1]^d$. More precisely, the NN $h^{(1,m,d)}$ has $m \cdot d$ many neurons in the single hidden layer, who are partitioned into $m$ groups. The weights are chosen such that the output of the neurons in one group depends only on one input variable and divides the interval $[0,1]$ into $m$ subintervals of equal length, each of which is then mapped to the unit interval $[0,1]$ by the output neuron. Figure 2 illustrates this construction. In Appendix 5.2.1 or in Montúfar et al. (2014), the reader can find an explicit construction of $h^{(1,m,d)}$.

The map $h^{(1,m,d)}$ identifies only $O(m^d)$ many cubes with each other. To subdivide the input space into exponentially many cubes and map them to the unit cube, we need a deep neural network. For this purpose, we utilize a vector $\mathbf{m}$ of folding factors instead of a single number $m$. Let $\mathbf{m} = (m_1, \ldots, m_L) \in \mathbb{N}^L$ with $m_\ell$ even for all $\ell \in [L]$ and define the neural network $h^{(L,\mathbf{m},d)}$ with $L$ hidden layers as $h^{(L,\mathbf{m},d)} = h^{(1,m_L,d)} \circ \cdots \circ h^{(1,m_1,d)}$. Since each of the $m_1^d$ cubes that results from the subdivision by the first layer is mapped back to $[0,1]^d$, each cube is subdivided again into $m_2^d$ cubes by the subsequent layer. Thus, after $L$ such layers, we obtain a subdivision of the input space into $\left(\prod_{\ell=1}^L m_\ell\right)^d$ cubes.

In the following, we define variables that are fixed but arbitrary: $L \in \mathbb{N}$, $\mathbf{m} = (m_1, \ldots, m_L) \in \mathbb{N}^L$ and $M = \left(\prod_{\ell=1}^L m_\ell\right)$ with $m_\ell > 1$ even for all $\ell \in [L]$. The following lemma states that $h^{(L,\mathbf{m},d)}$ actually enjoys the aforementioned properties.

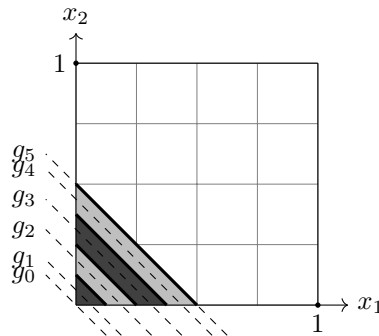

Figure 3: Illustration of the preimage of the map $\hat{g}^{(4,2)}$ in $[0,1]^2$, where the darkgray regions correspond to $\hat{g}^{-1}((0,\infty))$ and the lightgray regions to $\hat{g}-1((-\infty,0))$.

Figure 4: Illustration of the preimage of the map $g^{(4,2)}$ in $[0,1]^2$.

**Lemma 4.** *(cf. ([Montúfar et al., 2014](#))) Let $d \in \mathbb{N}$, then:*

1. $h^{(L,\mathbf{m},d)}(W^{(L,\mathbf{m},d)}_{(i_1,\ldots,i_d)}) = [0,1]^d$

2. $\pi_j \circ h^{(L,\mathbf{m},d)}_{|W^{(L,\mathbf{m},d)}_{(i_1,\ldots,i_d)}}(x_1,\ldots,x_d) = \begin{cases} M \cdot x_j - (i_j - 1) & i_j \ odd \\ -M \cdot x_j + i_j & i_j \ even \end{cases}$

*for all $(i_1,\ldots,i_d) \in [M]^d$.*

We now define cutting points as the points that are mapped to the point $(1,1,1,\ldots,1,0)$ by the map $h^{(L,\mathbf{m},d)}$ since they will play a central role in counting the annuli in the sublevel set of $F$.

**Definition 2.** *We call a point $x \in [0,1]^d$ a cutting point if it has coordinates of the form $x_i = \frac{x_i'}{M}$ for all $i \in \{1,\ldots,d\}$, where the $x_i'$ are odd integers for $1 \le i \le d-1$ and $x_d'$ is an even integer.*

Next, for $w \ge 2$, we build a 1-hidden layer neural network $\hat{g}^{(w,d)} \colon \mathbb{R}^d \to \mathbb{R}$ that cuts the d-dimensional unit cube into $w$ pieces such that $\hat{g}^{(w,d)}$ maps the pieces alternatingly to $\mathbb{R}_{>0}$ and $\mathbb{R}_{<0}$, respectively. We omit the indices $w$ and $d$ whenever they are clear from the context.

In order to build the neural network, we fix $w$ and $d$ and define the maps $\hat{g}_q \colon \mathbb{R}^d \to \mathbb{R}$, $q = 0,\ldots,w+1$ by

$$\hat{g}_q(x) = \begin{cases} \max\{0, \mathbf{1}^T x\} & q = 0 \\ \max\{0, \mathbf{1}^T x - 1\} & q = w+1 \\ \max\{0, 2(\mathbf{1}^T x - (2q-1)/4w)\} & \text{else} \end{cases}$$

and let $\hat{g} \colon \mathbb{R}^d \to \mathbb{R}$ be given by

$$\hat{g}(x) = \sum_{q=0}^{w+1} (-1)^q \cdot \hat{g}_q(x).$$

Later in this section, we will iteratively construct $k$-annuli in the sublevel set of $F$ for all $k \in [d-1]$. In order to ensure that these annuli are disjoint, it is convenient to place them around the cutting points. To achieve this, we mirror the map $\hat{g}$ before precomposing it with $h$. The mirroring transformation that maps the origin to the point $(1,\ldots,1,0)$ is an affine map $t \colon [0,1]^d \to [0,1]^d$ defined by $t(x_1, x_2, \ldots, x_d) = (1 - x_1, 1 - x_2, \ldots, 1 - x_{d-1}, x_d)$. We define the neural network $g = \hat{g} \circ t$ as the consecutive execution of $\hat{g}$ and $t$.

**Lemma 5.** *Let $d,w \in \mathbb{N}$ and*

$$R_q = \{x \in [0,1]^d : \frac{q}{2w} < \|(1,1,\ldots,1,0) - x\|_1 < \frac{q+1}{2w}\}.$$

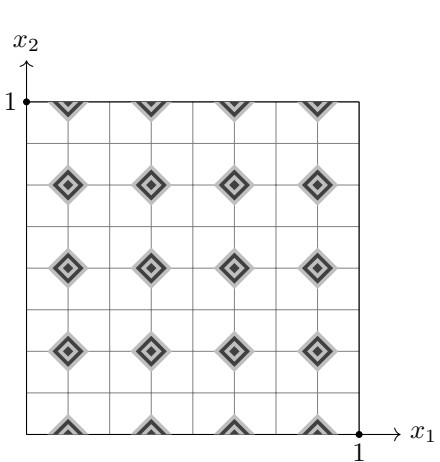
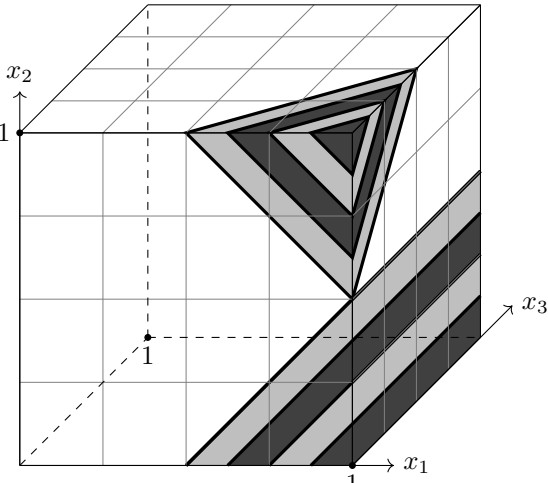

Figure 5: Illustration of the preimage of the composition $g^{(4,2)} \circ h^{(3,2,2)}$.

Figure 6: Illustration of the preimage of $f^{(4,4)} = g^{(4,3)} + g^{(4,2)} \circ p_2$.

Then there exists a 1-hidden layer neural network $g^{(w,d)} \colon \mathbb{R}^d \to \mathbb{R}$ of width $w + 2$ such that

$$\mathrm{sgn}(g^{(w,d)}(R_q)) = (-1)^q \; \forall q = 0, \ldots, w - 1$$

and $g^{(w,d)}(x) = 0$ for all $x \in [0,1]^d$ with $\|(1,1,\ldots,1,0) - x\|_1 \geq \frac{1}{2}$.

Lemma 20 in the appendix characterizes the regions around cutting points that admit positive respectively negative values under the map $g^{(w,d)} \circ h^{(L,\mathbf{m},d)}$. We focus on the regions that admit negative values, i.e., the space $Y_{d,w} := (g^{(w,d)} \circ h^{(L,\mathbf{m},d)})^{-1}((-\infty, 0))$ and observe that we obtain $d$-dimensional $(d-1)$-annuli around each cutting point.

Combining Lemma 20 and further observations about the number of cutting points (cf. Observation 21 in the appendix), we can finally describe $Y_{d,w}$ as a topological space.

**Proposition 6.** *The space $Y_{d,w}$ is homeomorphic to the disjoint union of $p_d = \frac{M^{(d-1)}}{2^{d-1}} \cdot \left(\frac{M}{2} - 1\right) \cdot \left\lceil \frac{w}{2} \right\rceil$ many $(d-1)$-annuli and $p'_d = \frac{M^{(d-1)}}{2^{d-2}} \cdot \left\lceil \frac{w}{2} \right\rceil$ many disks, that is,*

$$Y_{d,w} \cong \coprod_{k=1}^{p_d} (S^{d-1} \times [0,1]) \sqcup \coprod_{k=1}^{p'_d} D^d.$$

In order to obtain exponentially many $k$-annuli for all $k \in [d-1]$, we follow a recursive approach: At each step, we start with a $k$-dimensional space that has exponentially many $j$-annuli for all $j \in [k-1]$. We then cross this space with the interval $[0,1]$, transforming the $k$-dimensional $j$-annuli into $(k+1)$-dimensional $j$-annuli. Finally, we "carve" $(k+1)$-dimensional $k$-annuli in this newly formed product space. To allow us flexibility with respect to the numbers of annuli carved in different dimensions, we fix an arbitrary vector $\mathbf{w} = (w_1, \ldots, w_{d-1}) \in \mathbb{N}^{d-1}$ such that $\sum_{i=1}^{d-1}(w_i + 2) = n_{L+1}$. We iteratively define the 1-hidden layer neural network $f^{(w_1,\ldots,w_{k-1})} \colon \mathbb{R}^k \to \mathbb{R}$ of width $n_{L+1}$ by $f^{(w_1)} = g^{(w_1,2)}$ and

$$f^{(w_1,\ldots,w_{k-1})} = f^{(w_1,\ldots,w_{k-2})} \circ p_{k-1} + g^{(w_{k-1},k)}$$

for $k \leq d$. Roughly speaking, the following lemma states that the carving map does not interfere with the other maps, i.e., there is enough space in the unit cubes to place the $k$-annuli after having placed all $k'$-annuli ($k' < k$) in the same, inductive manner.

**Lemma 7.** *For $k \leq d$ and $\mathbf{w} = (w_1, \ldots, w_{d-1}) \in \mathbb{N}^{d-1}$ it holds that*

1. $f^{(w_1,\ldots,w_{k-2})} \circ p_{k-1}(x) \neq 0 \implies g^{(w_{k-1},k)}(x) = 0$ *and*

2. $g^{(w_{k-1},k)}(x) \neq 0 \implies f^{(w_1,\ldots,w_{k-2})} \circ p_{k-1}(x) = 0$

for all $x \in [0,1]^k$.

Using Lemma 7 and the fact that the folding maps $h^{(L,\mathbf{m},k)}$ are compatible with projections (cf. Lemma 22 in Appendix), we can make sure that we can construct the cuts iteratively so that we obtain $k$-annuli for every $k \in [d-1]$, which is stated in the following lemma.

**Lemma 8.** *For $2 \leq k \leq d$, the space $X_k := (f^{(w_1,\ldots,w_{k-1})} \circ h^{(L,\mathbf{m},k)})^{-1}((-\infty,0))$ satisfies*
$$X_k = (X_{k-1} \times [0,1]) \sqcup Y_{k,w}$$
*with $X_1 := \emptyset$.*

Lemma 8, Proposition 6 and the disjoint union axiom (Proposition 15 in Appendix 5.1.2) allow us to compute the Betti numbers of the decision region of $F := f^{(w_1,\ldots,w_{d-1})} \circ h^{(L,\mathbf{m},d)}$ as stated in Theorem 24 in the Appendix. One can easily generalize this statement by rounding down the widths $n_1,\ldots,n_L$ to the nearest even multiple of $d$:

**Theorem 9.** *Given an architecture $A = (d, n_1, \ldots, n_L, 1)$ with $n_\ell \geq 2d$ for all $\ell \in [L]$ and numbers $w_1, \ldots, w_{d-1} \in \mathbb{N}$ such that $\sum_{k=1}^{d-1}(w_k + 2) = n_L$, there is a neural network $F \in \mathcal{F}_A$ such that*

(i) $\beta_0(F) = \sum_{k=2}^{d} \frac{M^{(k-1)}}{2^{k-1}} \cdot \left(\frac{M}{2} + 1\right) \cdot \left\lceil \frac{w_k}{2} \right\rceil$

(ii) $\beta_k(F) = \frac{M^{(k-1)}}{2^{k-1}} \cdot \left(\frac{M}{2} - 1\right) \cdot \left\lceil \frac{w_{k-1}}{2} \right\rceil$ *for $0 < k < d$,*

*where $M = \prod_{\ell=1}^{L-1} 2 \cdot \lfloor \frac{n_\ell}{2d} \rfloor$.*

The special case $\lfloor \frac{w_1}{2} \rfloor = \ldots = \lfloor \frac{w_d}{2} \rfloor$ corresponds precisely to Corollary 1.

In order to obtain the lower bound we choose the weights explicitly, but the construction is robust to small perturbations. Basically this relies on the fact that since we have finitely many linear regions and no hyperplanes of non-linearity introduced at different applications of the ReLU function that coincide, one can perturb the weights slightly, such that the combinatorial structure of the polyhedral complex is preserved. From this we easily conclude the maintenance of the existence of all the annuli. In fact, if we denote by $\Phi \colon \mathbb{R}^D \to C(\mathbb{R}^d)$ the map that assigns a vector of weights to the function computed by the ReLU neural network with this weights, in Section 5.3 we prove the following:

**Proposition 10.** *There is an open set $U \subseteq \mathbb{R}^D$ in the parameter space of the architecture $(d, m \cdot d, \ldots, m \cdot d, w, 1)$ such that $\Phi(u)$ restricted to the unit cube has at least the same topological expressivity as $F$ in Theorem 24 for all $u \in U$.*

As mentioned previously, the sum of Betti numbers, the notion of topological expressivity used in Bianchini and Scarselli (2014), does not provide us with an understanding of holes of different dimensions. On the other hand, our bounds are clearly an extension of this result. In addition, the dimension-wise lower bound allows further implications, one of them being a lower bound on the *Euler characteristic*, which is the alternating sum $\chi(X) = \sum_{k=1}^{d} \beta_k(X)$ of the Betti numbers.

**Corollary 11.** *Let $A$ be the architecture as in Theorem 24, then there is a ReLU NN $F \colon \mathbb{R}^d \mapsto \mathbb{R}$ with architecture $A$ such that the space $X_d := F^{-1}((-\infty,0))$ satisfies $\chi(X_d) \in \Omega\left(M^d \cdot \sum_{i=1}^{d-1} w_i\right)$, where $\chi(X_d)$ denotes the Euler characteristic of the space $X_d$.*

## 3 UPPER BOUND

In this section we derive an upper bound for $\beta_k(F)$ for a ReLU neural network $F \colon \mathbb{R}^d \to \mathbb{R}$ for all $k \in [d-1]$, showing that they are polynomially bounded in the width using an upper bound on the linear regions of $F$. A linear region $R$ of $F$ contains at most one convex polyhedral subspace where $F$ takes on exclusively nonnegative function values. Intuitively, every such polyhedral subspace can be in the interior of at most one $d$-dimensional hole of the sublevel set $F^{-1}((-\infty,0))$ and thus the number of linear regions is an upper bound for $\beta_{d-1}(F)$. In the following proposition we will formalize this intuition and generalize it to $\beta_k(F)$ for all $k \in [d-1]_0$.

**Proposition 3.** *Let $F\colon \mathbb{R}^d \to \mathbb{R}$ be a neural network of architecture $(d, n_1, \ldots, n_L, 1)$. Then it holds that $\beta_0(F) \leq \sum_{(j_1,\ldots,j_L)\in J} \prod_{\ell=1}^{L} \binom{n_\ell}{j_\ell}$ and for all $k \in [d-1]$ that*

$$\beta_k(F) \leq \binom{\sum_{(j_1,\ldots,j_L)\in J} \prod_{\ell=1}^{L} \binom{n_\ell}{j_\ell}}{d-k},$$

*where $J = \big\{ (j_1,\ldots,j_L) \in \mathbb{Z}^L : 0 \leq j_\ell \leq \min\{d, n_1 - j_1, \ldots, n_{\ell-1} - j_{\ell-1}\} \text{ for all } \ell = 1, \ldots, L\big\}$.*

*Proof sketch.* Theorem 1 in (Serra et al., 2017) states that $F$ has at most $\sum_{(j_1,\ldots,j_L)\in J} \prod_{l=1}^{L} \binom{n_l}{j_l}$ linear regions. In Section 5.4 we will provide a formal proof for the statement that we sketch here. Let $\mathcal{P}$ be the canonical polyhedral complex of $F$, i.e, $F$ is affine linear on all polyhedra in $\mathcal{P}$ (c.f Appendix 12). For every $k \in [d]$ we will define a $\mathcal{P}_k^-$ as a subcomplex of a subdivison of the $(k+1)$-skeleton of $\mathcal{P}$ such that $F$ takes on exclusively negative respectively nonnegative values on the $k+1$-dimensional polyhedra of $\mathcal{P}_k^-$ in such a way that $\beta_k(\mathcal{P}_k^-) = \beta_k(F)$. We then proceed by showing the chain of inequalities $\beta_k(\mathcal{P}_k^-) \leq \#\mathcal{P}(k+1) \leq \binom{\sum_{(j_1,\ldots,j_L)\in J} \prod_{\ell=1}^{L} \binom{n_\ell}{j_\ell}}{d-k}$ using cellular homology and polyhedral geometry, where $\mathcal{P}(k+1) \subseteq \mathcal{P}$ is the set of $k+1$-dimensional polyhedra in $\mathcal{P}$. This concludes the proof, since it also holds that $\beta_0(\mathcal{P}^-) \leq \#\mathcal{P}_d^-(d) \leq \mathcal{P}(d) = \sum_{(j_1,\ldots,j_L)\in J} \prod_{l=1}^{L} \binom{n_l}{j_l}$,

$\square$

This implies that the upper bound is polynomial in the width:

**Corollary 12.** *Let $F\colon \mathbb{R}^d \to \mathbb{R}$ be a neural network of architecture $(d, n, \ldots, n, 1)$ and depth $L$, then $\beta_k(F^{-1}((-\infty, 0))) \in O(n^{L \cdot d \cdot (d-k)})$ for $k \in [d-1]$ and $\beta_0(F^{-1}((-\infty, 0))) \in O(n^{L \cdot d})$.*

## 4 CONCLUSION, LIMITATIONS AND OUTLOOK

Since it is widely accepted that data sets often have nontrivial topologies, investigating a neural network's ability to capture topological properties, as characterized by all Betti numbers, is an exciting and essential question that yields insight into the nature of ReLU networks. In an attempt to shed light on this question, we proved lower and upper bounds for the topological expressivity of ReLU neural networks with a given architecture. Our bounds give a rough estimate on how the architecture needs to be in order to be at least theoretically able to capture the topological complexity of the data set in these dimensions; in particular, in the first few dimensions where Betti numbers are computable in practice.

As a byproduct of our analysis we saw that two hidden layers are sufficient to increase the topological expressivity as much as we want at the expense of an increased width. Even though Betti numbers are a common complexity measure for topological spaces in data analysis, they only provide a coarse classification, i.e., two spaces can have the same Betti numbers but still look very different. Although there are finer topological invariants such as cohomology rings or homotopy groups, from a computational point of view, Betti numbers are a good trade-off between the ability to capture differences of spaces and tractability. Nevertheless, it might be interesting to find further topological or geometrical invariants to investigate the expressivity of neural networks in the setting of classification tasks.

Even though our lower bounds apply under certain restrictions of neural network architecture, this does not pose a big limitation for our purposes. Since our results are of a theoretical and mostly asymptotic nature, a constant factor (in the hidden layers resp. in the last hidden layer) is negligible. Besides, since our layers merely consists of many small layers put in parallel, one could also concatenate the layers in order to achieve a smaller width maintaining all the asymptotic results.

It seems straightforward that the construction in Section 2 can be adapted to neural networks with sigmoidal activation functions in a "smoothed" way. Therefore we conjecture that the same lower bound holds for the topological expressivity of neural networks with sigmoidal activation function, which would generalise the lower bound for the zeroth Betti number given in Bianchini and Scarselli (2014) to all Betti numbers.

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

## 5 APPENDIX

### 5.1 MATHEMATICAL BACKGROUND

#### 5.1.1 POLYHEDRAL GEOMETRY

**Definition 3** (Polyhedral complex). *A collection of polyhedra $\mathcal{P}$ is called a* polyhedral complex *if*

1. *Every face $F$ of any polyhedra $P \in \mathcal{P}$ is also in $\mathcal{P}$ and*

2. *it holds that $P \cap Q \in \mathcal{P}$ for all $P, Q \in \mathcal{P}$.*

*There is a poset structure given on $\mathcal{P}$ by $Q \preceq P :\iff Q$ is a face of $P$ and we call $(\mathcal{P}, \preceq)$ the* face poset *of the polyhedral complex. Furthermore we define*

$$\mathcal{P}_k := \{k - dimensional\ polyehdra\ in\ \mathcal{P}\}.$$

*Note that for any polyhedron, the set of all its faces forms a polyhedral complex .*

**Definition 4** (Isomorphisms of polyhedral complexes and polytopes)**.** *Let $\mathcal{P}$ and $\mathcal{Q}$ be polyhedral complexes. A map $\varphi \colon \mathcal{P} \to \mathcal{Q}$ is called an isomorphism if it is an isomorphism of the face posets of $\mathcal{P}$ and $\mathcal{Q}$ and it holds that $\dim(\varphi(P)) = \dim(P)$ for all $P \in \mathcal{P}$.*

*If $P$ and $Q$ are polytopes we call a map $\varphi \colon P \to Q$ an isomorphism if it is an isomorphism when considering $P$ and $Q$ as polyhedral complexes.*

**Definition 5.** *We call $\varphi \colon \mathcal{P} \to \mathcal{Q}$ an $\varepsilon$-isomorphism if it is an isomorphism (of polyhedral complexes) and it holds that $\|\varphi(v) - v\|_2 < \varepsilon$ for all $v \in \mathcal{P}_0$.*

**Definition 6.** *Let $x \mapsto a^T x + b$ be an affine linear map and $H(a, b) := \{x \in \mathbb{R}^d \mid a^T x + b = 0\}$ the hyperplane given by the kernel. Then we denote the corresponding half-spaces by*

$$H^1(a, b) := \{x \in \mathbb{R}^d \mid a^T x \geq b\},$$

$$H^{-1}(a, b) := \{x \in \mathbb{R}^d \mid a^T x \leq b\}.$$

*We will also use the notation $H^0(a, b) := H(a, b)$. We will simply write $H^s$ for $H^s(a, b)$ whenever $a$ and $b$ are clear from the context.*

**Lemma 13.** *Let $P \subseteq \mathbb{R}^d$ be a polytope, $a \in \mathbb{R}^d$ and $b \in \mathbb{R}$ such that $P_0 \cap H(a, b) = \emptyset$. Then for all $\varepsilon > 0$, there is a $\delta > 0$ such that for all $(a', b') \in B_\delta^{d+1}((a, b))$ there are $\varepsilon$-isomorphisms*

$$\psi^s \colon P \cap H^s(a, b) \to P \cap H^s(a', b')$$

*for $s \in \{-1, 0, 1\}$. Furthermore it holds that $P_0 \cap H(a', b') = \emptyset$.*

*Proof.* Let $e \in P_1$ and $\mathbb{R}_e := \mathrm{Aff}(e)$ be the affine space spanned by $e$. First, assume that $\mathbb{R}_e \cap H(a, b) \neq \emptyset$. Since $H(a, b) \cap P_0 = \emptyset$ we know that $\mathbb{R}_e \cap H(a, b) = \{v_e^{(a,b)}\}$ with $v_e^{(a,b)} \in e^\circ$ or $v_e^{(a,b)} \in \mathbb{R}_e \setminus e$, where $e^\circ$ denotes the relative interior of $e$. Let

$$\varepsilon_e := \begin{cases} \min\{\varepsilon, \frac{1}{2} \inf_{y \in e^\circ} \|y - v_e^{(a,b)}\|_\infty\} & v_e^{(a,b)} \in e^\circ \\ \min\{\varepsilon, \frac{1}{2} \inf_{y \in \mathbb{R}_e \setminus e} \|y - v_e^{(a,b)}\|_\infty\} & v_e^{(a,b)} \in \mathbb{R}_e \setminus e \end{cases}$$

It is easily verified that the map $(c, d) \mapsto H(c, d) \cap \mathbb{R}_e$ is locally continuous around $(a, b)$ and hence there is a $\delta_e > 0$ such that $\|(a, b) - (a', b')\| < \delta_e$ implies that $\|v_e^{(a,b)} - v_e^{(a',b')}\|_\infty < \varepsilon_e$ for all $e \in P_1$. On the other hand, if $\mathbb{R}_e \cap H(a, b) = \emptyset$, then there is a $\delta_e > 0$ such that $e^\circ \cap H(a', b') = \emptyset$. Let $\delta := \min_{e \in P_1} \delta_e$. Note that $(P \cap H(a, b))_0 = \{v_e^{(a,b)} \mid v_e^{(a,b)} \in e^\circ\}$ and $(P \cap H(a', b'))_0 = \{v_e^{(a',b')} \mid v_e^{(a',b')} \in e^\circ\}$ and hence $f(v_e^{(a,b)}) := v_e^{(a',b')}$ defines a bijection $f \colon (P \cap H)_0 \to (P \cap H')_0$ for $(a', b') \in B_\delta^{d+1}((a, b))$. Let $F$ be a face of $P \cap H(a, b)$, then $F = F' \cap H(a, b)$ for some face $F'$ of $P$ and furthermore $F = \mathrm{conv}(\{v_e^{(a,b)} \cap e \mid e \preceq F\})$. It now easily follows by induction on the dimension of the face $F$ that $F$ is isomorphic to $\mathrm{conv}(\{v_e^{(a',b')} \cap e \mid e \preceq F\})$ and therefore in particular that $P \cap H(a, b)$ is isomorphic to $P \cap H(a', b')$. We can extend $f$ to a bijection $f \colon (P \cap H^s(a, b))_0 \to (P \cap H^s(a', b'))_0$ by $f(v) = v$ for all $v \in P_0 \cap H^s(a, b)$ and by the same arguments we obtain that $P \cap H^s(a, b)$ is isomorphic to $P \cap H^s(a', b')$ for $s \in \{-1, 1\}$. $\qquad\square$

**Lemma 14.** *Let $P \subseteq \mathbb{R}^d$ be a polytope, $H = \{x \in \mathbb{R}^d \mid a^T x = b\}$ be a hyperplane. If $P_0 \cap H = \emptyset$ then for all $\varepsilon > 0$ there is a $\delta > 0$ such that for all polytopes $Q \subseteq \mathbb{R}^d$ and all $\delta$-isomorphisms $\varphi \colon P \to Q$ there are $\varepsilon$-isomorphisms*

$$\gamma^s \colon P \cap H^s \to Q \cap H^s$$

*for $s \in \{-1, 0, 1\}$ and furthermore it holds that $P_0 \cap H' = \emptyset$.*

*Proof.* Let $e \in P_1$ and let $\partial e = \{u, v\}$. We will also use the notation $e = uv$ and define

$$\delta_e := \min\{\varepsilon, \frac{1}{2}\inf_{y \in H}\|y - v\|_\infty, \frac{1}{2}\inf_{y \in H}\|y - u\|_\infty\}$$

and $\delta := \min_{e \in P_1} \delta_e$. Since $P_0 \cap H = \emptyset$ it holds that $\delta > 0$. Let $\varphi \colon P \to Q$ be a $\delta$-isomorphism. Then it holds $H \cap uv = \{v_e\} \neq \emptyset$ if and only if $H \cap \varphi(u)\varphi(v) = \{v_{\varphi(e)}\} \neq \emptyset$. Note that $(P \cap H)_0 = \{v_e \mid H \cap e \neq \emptyset\}$ and $(Q \cap H)_0 = \{v_{\varphi(e)} \mid H \cap \varphi(e) \neq \emptyset\}$ and hence $f(v_e) := v_{\varphi(e)}$ defines a bijection $f \colon (P \cap H)_0 \to (Q \cap H)_0$. The remaining proof is equivalent to the proof of Lemma 13.

$\square$

### 5.1.2 TOPOLOGY

In the following, we summarize background knowledge necessary for our purposes that the reader may not have been acquainted with. The content of this subsection can also be found in many algebraic topology textbooks such as Hatcher (2002, Chapter 2).

First, we recall two well-known constructions in topology that yield well-behaved, yet more complex topological spaces.

**Definition 7.** *For two topological spaces $X$ and $Y$, the space $X \sqcup Y$ denotes the* disjoint union *of $X$ and $Y$ endowed with the disjoint union topology. Similarly for an arbitrary index set $I$, the set $\bigsqcup_{i \in I} X_i$ denotes the disjoint union of the topological spaces $X_i$ for $i \in I$. If $I$ is a finite set, i.e., $I = \{1, \ldots, q\}$ for a suitable $q \in N$, we also denote this space by $\bigsqcup_{i=1}^{q} X_i$.*

We also create *product spaces*: For two topological spaces $X$ and $Y$, the product space is the Cartesian product $X \times Y$ endowed with the product topology. Even though it is possible to extend this definition to infinite families of topological spaces as well, this will not be needed for our purposes.

Next, we introduce the notion of homology by giving a sketch of the construction of homology groups.

Let $X$ be a topological space and

$$\Delta_n = \left\{\sum_{i=0}^{n}\theta_i x_i \colon x \in \mathbb{R}^n, \sum_{i=0}^{n}\theta_i = 1, \theta_i \geq 0 \text{ for all } i = 0, \ldots, n\right\}$$

denote the standard $n$-simplex. Note that the standard $n$-simplex is the convex combination of $n+1$ points $\{p_0, \ldots, p_n\}$. Taking the convex combination of an $n$-subset $\{p_0, \ldots, p_n\} \setminus \{p_i\}$ of these points, one obtains a subspace homeomorphic to the standard $n-1$-simplex, which we call an *$i$-th $n$-face* of the simplex.

The $\mathbb{Z}$-module $C_n$, the group of *$n$-chains*, is defined as the free abelian group generated by continuous maps $\sigma \colon \Delta_n \to X$, called *simplices*. The inclusion $\iota_i \colon \Delta_{n-1} \hookrightarrow \Delta_n$ induces $n-1$-simplices $\sigma_i := \sigma \circ \iota_i \colon \Delta_{n-1} \to X$ by the inclusion of the $i$-th $n$-face into the standard simplex.

The map $\partial_n \colon C_n \to C_{n-1}$, which we call the *boundary map*, is constructed as $\sigma \mapsto \sum_{i=1}^{n}(-1)^i\sigma_i$ on the generators and by linear extension elsewhere. This yields a *chain complex*

$$\ldots \to C_{n+1} \xrightarrow{\partial_{n+1}} C_n \xrightarrow{\partial_n} C_{n-1} \to \ldots,$$

where we have $\partial_{n-1} \circ \partial_n = 0$ for all $n$. Therefore, we can define the *$n$-th singular homology group* as

$$H_n(X) := {}^{\ker(\partial_n)}\!/\!_{\mathrm{im}(\partial_{n+1})}.$$

We list some well-known properties of (singular) homology groups that will be used in our constructions.

**Proposition 15.** *Let $n \in \mathbb{N}$.*

1. *(Disjoint union axiom, implication) For any index set $I$ and topological spaces $X_i$ for $i \in I$, it holds that $H_n \left( \bigsqcup_{i \in I} X_i \right) \cong \bigoplus_{i \in I} H_n(X_i)$.*

2. *(Homotopy invariance axiom, special case) Let $X$ be a topological space and $n, d \in \mathbb{N}$, then it holds that $H_n(X \times D^d) \cong H_n(X)$.*

3. *(Dimension axiom) $H_n(D^d) = \begin{cases} \mathbb{Z}, & n = 0 \\ 0 & else \end{cases}$*

4. *$H_n(S^d) = \begin{cases} \mathbb{Z} \oplus \mathbb{Z} & n = d = 0 \\ \mathbb{Z}, & n = d \neq 0 \text{ or } d \neq n = 0 \\ 0 & else \end{cases}$*

**Observation 16.** *Using Proposition 15 and given definitions, one can immediately calculate the homology groups of a $d$-dimensional $k$-annuli:*

$$H_n(S^k \times D^{d-k}) = H_n(S^k) = \begin{cases} \mathbb{Z} \oplus \mathbb{Z} & n = k = 0 \\ \mathbb{Z}, & n = k \neq 0 \text{ or } k \neq n = 0 \\ 0 & else \end{cases}$$

To ease our computations for upper bounds, we deviate to another homology theory called *cellular homology* which is defined on a special class of topological spaces called *CW-complexes*.

**Definition 8.** *A Hausdorff space $X$ with a filtration $\emptyset = X_{-1} \subseteq X_0 \subseteq \ldots \subseteq \bigcup_{i=1}^{d} X_d = X$ is a $d$-dimensional finite CW complex if the following axioms hold:*

(i) *A subset $A \subseteq X$ is closed in $X$ if and only if $A \cap X_i$ is closed in $X_i$ for all $i \in [d]_0$.*

(ii) *The spaces $X_i$ in the filtration are each called $i$-skeleton. The $i$-skeleton is recursively obtained from $X_{i-1}$ by attaching cells, i.e. we have pushout maps of the form*

$$\begin{array}{ccc} \bigsqcup_{j \in I_i} S^{i-1} & \xrightarrow{\bigsqcup_{j \in I_i} q_i^j} & X_{i-1} \\ \downarrow & & \downarrow \\ \bigsqcup_{j \in I_i} D^i & \xrightarrow{\bigsqcup_{j \in I_i} Q_i^j} & X_i \end{array}$$

*for finite index sets $I_i$ for $i \in [d]_0$. The maps $q_i^j$ are called* attaching maps *and the maps $Q_i^j$ are called* characteristic maps.

*For $i \in [d]_0$, the set of path components of $X_i \setminus X_{i-1}$ is called the set of* open $i$-cells. *The set of closures of open $i$-cells are called* closed $i$-cells. *We almost always make use of closed $i$-cells and therefore refer to them simply as $i$-cells.*

The non-expert can understand a pushout map as one that simply glues the boundary of an $i$-dimensional cell (that is, a topological disk/polyhedron of dimension $i$) onto the $(i-1)$-skeleton $X_i$. Here, the choice of the attaching maps define the commutative pushout diagram above, while the characteristic maps are those that are "uniquely" defined by the attaching maps "in a natural way". Figure 7 illustrates the above definition.

A more general definition of CW complexes allow an infinte dimension of cells as well as an (arbitrarily indexed) infinite number of cells in every dimension. For our results, it is sufficient to restrict to our definition, as we will restrict to a CW complex that is a compact topological space without loss of generality:

**Lemma 17.** *A CW-complex is finite if and only if it is compact.*

One can naturally endow polyhedral complexes with CW-structures by defining the $i$-cells as the $i$-facets (therefore uniquely defining the filtration), and the attaching maps as those that include each face into the $i$-skeleton of the CW complex for each $i$. This way, the poset structure is compatible

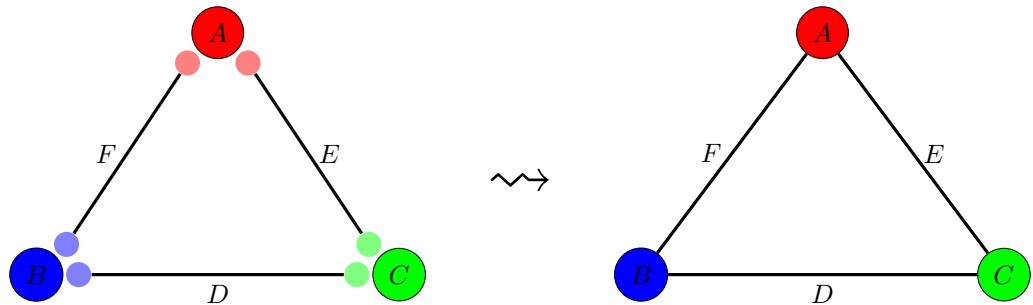

Figure 7: A CW-complex $X$ of dimension 1 that is homeomorphic to $S^1$. The darker shaded points constitute the 0-cells, i.e., we have $X_0 = \{A, B, C\}$. The line segments on the left are the 1-cells. The triangle on the right illustrates $X = X_1$. The lighter shades of colors indicate the attaching maps $q_1^j : S^0 \to \{A, B, C\}$ for $j \in \{D, E, F\}$.

with the characteristic maps as well (because faces of polyhedra lie on their topological boundary and the attaching maps $q_i^j$ are injective in this case).

Our motivation to endow polyhedral complexes with CW-structures is to use *cellular homology*, which, given the natural CW-structure of a polyhedral complex, allows to compute homology groups conveniently. Among other advantages, we will rely on cellular homology for an induction on the number of polyhedra.

**Definition 9.** *Let $X$ be a finite CW-complex. The* cellular chain complex $(C_i)_{i \in \mathbb{N}}$ *of $X$ is given by free abelian groups $C_i \cong \mathbb{Z}^{|I_i|}$ that are generated by the $i$-cells of $X$. The boundary maps, which are given by a construction using attachment maps in the general case, can be greatly simplified for our purposes: The boundary map $\partial_i : C_i \to C_{i-1}$ is defined by the incidence matrix $\Delta \in \mathbb{Z}^{|I_{i-1}| \times |I_i|}$ between $i$ and $(i-1)$-cells, that is, it is given by entries*

$$\delta_{jk} = \begin{cases} 1 & \text{the } (i-1)\text{-cell } j \text{ lies in the boundary of the } i\text{-cell } k \\ 0 & \text{else.} \end{cases}$$

*The $i$-th cellular homology $H_i^{cell}(X)$ of $X$ is defined by the homology of the cellular chain complex $(C_i)_{i \in \mathbb{N}}$, that is, we have*

$$H_i^{cell}(X) = {}^{\ker(\partial_i)}\!/_{\text{im}(\partial_{i-1})}.$$

It is well-known that on CW-complexes, cellular homology groups coincide with singular homology groups. Therefore, we may make use of cellular homology groups in order to compute Betti numbers.

## 5.2 PROOFS AND ADDITIONAL STATEMENTS

In this section we provide formal proofs for the statements made in sections 2 and 3 and additional lemmata that are used in these proofs. For the sake of completeness, we also recall the statements we prove.

### 5.2.1 PROOF OF LEMMA 4

**Definition 10.** *We define the one hidden layer ReLU neural network $h^{(1,m,d)}$ in the following way: The neurons $\{v_{i,j}\}_{i=0,\dots,m-1, j=1,\dots,d}$ in the hidden layer are given by:*

- $v_{0,j}(x) = \max\{0, mx_j\}, j = 1, \dots, d$

- $v_{i,j}(x) = \max\{0, 2m(x_j - i/m)\}, j = 1, \dots, d \; i = 1, \dots, m-1$

*and the output neurons by: $h_j^{(1,m,d)}(x) = \sum_{i=0}^{m-1} (-1)^i \cdot v_{i,j}(x)$.*

**Lemma 18.** *Let $d, m \in \mathbb{N}$ with $m > 1$. Then*

1. $h^{(1,m,d)}(W^{(1,m,d)}_{(i_1,\ldots,i_d)}) = [0,1]^d$

2. $\pi_j \circ h^{(1,m,d)}_{|W^{(1,m,d)}_{(i_1,\ldots,i_d)}}(x_1,\ldots,x_d) = \begin{cases} m \cdot x_j - (i_j - 1) & i_j \text{ odd} \\ -m \cdot x_j + i_j & i_j \text{ even} \end{cases}$

*for all* $(i_1,\ldots,i_d) \in [m]^d$.

*Proof.* Throughout this proof we denote $W^{(1,m,d)}_{(i_1,\ldots,i_d)}$ by $W_{(i_1,\ldots,i_d)}$ and $h^{(1,m,d)}$ by $h$. We prove that $h$ satisfies the second property. The first property then follows immediately from the second since

$$\pi_j \circ h_{|W_{(i_1,\ldots,i_d)}}(W_{(i_1,\ldots,i_d)}) = \left[ m \cdot \frac{(i_j-1)}{m} - (i_j - 1), m \cdot \frac{i_j}{m} - (i_j - 1) \right] = [0,1]$$

if $i_j$ is odd and

$$\pi_j \circ h_{|W_{(i_1,\ldots,i_d)}}(W_{(i_1,\ldots,i_d)}) = \left[ m \cdot \frac{(i_j-1)}{m} + i_j, m \cdot \frac{i_j}{m} + i_j \right] = [0,1]$$

if $i_j$ is even.

Let $j \in \{1,\ldots,d\}$ and $x \in W_{(i_1,\ldots,i_d)}$, so in particular $x_j \in \left[\frac{(i_j-1)}{m}, \frac{i_j}{m}\right]$. Since $i \geq i_j$ implies $2m(x_j - i/m) \leq 0$, it follows that $v_{i,j}(x) = 0$ for all $i \geq i_j$. Similarly, $i < i_j$ implies $2m(x_j - i/m) \geq 0$, and therefore it follows that $v_{i,j}(x) = 2m(x_j - i/m)$ for all $i < i_j$. Hence

$$h_j(x) = \sum_{i=0}^{i_j-1} (-1)^i \cdot v_{i,j}(x) = mx_j + \sum_{i=1}^{i_j-1} (-1)^i \cdot 2m(x_j - i/m).$$

If $i_j$ is even, then

$$\begin{aligned} h_j(x) &= mx_j + \sum_{i=1}^{i_j/2-1} 2m(x_j - 2i/m) - \sum_{i=1}^{i_j/2} 2m(x_j - (2i-1)/m) \\ &= mx_j - 2(i_j/2 - 1) - 2m(x_j - (2i_j/2 - 1)/m) \\ &= mx_j - i_j + 2 - 2mx_j + 2i_j - 2 \\ &= -mx_j + i_j \end{aligned}$$

If $i_j$ is odd, then

$$\begin{aligned} h_j(x) &= mx_j + \sum_{i=1}^{(i_j-1)/2} 2m(x_j - 2i/m) - \sum_{i=1}^{(i_j-1)/2} 2m(x_j - (2i-1)/m) \\ &= mx_j - 2(i_j - 1/2) \\ &= mx_j - (i_j - 1). \end{aligned}$$

$\square$

**Lemma 4.** *(cf. (Montúfar et al., 2014)) Let $d \in \mathbb{N}$, then:*

1. $h^{(L,\mathbf{m},d)}(W^{(L,\mathbf{m},d)}_{(i_1,\ldots,i_d)}) = [0,1]^d$

2. $\pi_j \circ h^{(L,\mathbf{m},d)}_{|W^{(L,\mathbf{m},d)}_{(i_1,\ldots,i_d)}}(x_1,\ldots,x_d) = \begin{cases} M \cdot x_j - (i_j - 1) & i_j \text{ odd} \\ -M \cdot x_j + i_j & i_j \text{ even} \end{cases}$

*for all* $(i_1,\ldots,i_d) \in [M]^d$.

*Proof.* We apply induction over $L$. The base case has already been covered by Lemma 18. Assume that there exists a NN $h^{(L-1,(m_1,\ldots,m_{L-1}),d)}$ that satisfies the desired properties and define $h^{(L,\mathbf{m},d)} = h^{(1,m_L,d)} \circ h^{(L-1,(m_1,\ldots,m_{L-1}),d)}$. Let $j \in \{1,\ldots,d\}$ and $x \in W^{(L,\mathbf{m},d)}_{(i_1,\ldots,i_d)}$. Define $i^{(1)}_j := \left\lfloor \frac{m_L \cdot (i_j-1)}{M} \right\rfloor + 1$. It holds that $\left[\frac{(i_j-1)}{M}, \frac{i_j}{M}\right] \subset \left[\frac{(i^{(1)}_j-1)}{m_L}, \frac{i^{(1)}_j}{m_L}\right]$.

**Case 1:** $i_j^{(1)}$ odd. Then by Lemma 18:

$$h_j^{(1,m_L,d)}\left(\tfrac{(i_j-1)}{M}\right) = m_L \cdot \frac{(i_j-1)}{M} - (i_j^{(1)}-1) = \frac{m_L \cdot (i_j-1)}{M} - \left\lfloor \frac{m_L \cdot (i_j-1)}{M} \right\rfloor = \frac{(i_j-1) \bmod (M/m_L)}{(M/m_L)}$$

and

$$h_j^{(1,m_L,d)}\left(\tfrac{i_j}{M}\right) = m_L \cdot \frac{i_j}{M} - (i_j^{(1)}-1) = \frac{m_L \cdot (i_j)}{M} - \left\lfloor \frac{m_L \cdot (i_j-1)}{M} \right\rfloor = \frac{(i_j-1) \bmod (M/m_L)+1}{(M/m_L)}.$$

Define $i_j^{(L-1)} := ((i_j - 1) \bmod (M/m_L)) + 1.$ Then it holds that $h_j^{(1,m_L,d)}(x) \in \left[\frac{m_L \cdot (i_j^{(L-1)}-1)}{M}, \frac{m_L \cdot i_j^{(L-1)}}{M}\right]$. Moreover, $i_j^{(L-1)} = ((i_j - 1) \bmod (M/m_L)) + 1$ is odd if and only if $i_j$ is odd because $\frac{M}{m_L}$ is an even number.

**Case 1.i:** $i_j^{(L-1)}$ (and therefore $i_j$) is odd. Then follows with the induction hypothesis:

$$\begin{aligned}
h_j^{(L,\mathbf{m},d)}(x) &= h_j^{(L-1,(m_1,\dots,m_{L-1}),d)}(h_j^{(1,m_L,d)}(x)) \\
&= \frac{M}{m_L} \cdot (h_j^{(1,m_L,d)}(x)) - (i_j^{(L-1)} - 1) \\
&= \frac{M}{m_L} \cdot (m_L x - (i_j^{(1)} - 1)) - (i_j^{(L-1)} - 1) \\
&= Mx - \frac{M}{m_L} \cdot \left\lfloor \frac{m_L \cdot (i_j-1)}{M} \right\rfloor - ((i_j - 1) \bmod (M/m_L)) \\
&= Mx - (i_j - 1).
\end{aligned}$$

**Case 1.ii:** $i_j^{(L-1)}$ (and therefore $i_j$) is even. Then follows with the induction hypothesis:

$$\begin{aligned}
h_j^{(L,\mathbf{m},d)}(x) &= h_j^{(L-1,(m_1,\dots,m_{L-1},d)}(h_j^{(1,m_L,d)}(x)) \\
&= -\frac{M}{m_L} \cdot (h_j^{(1,m_L,d)}(x)) + i_j^{(L-1)} \\
&= -\frac{M}{m_L} \cdot (m_L x - (i_j^{(1)} - 1)) + i_j^{(L-1)} \\
&= -Mx + \frac{M}{m_L} \cdot \left\lfloor \frac{m_L \cdot (i_j-1)}{M} \right\rfloor + ((i_j - 1) \bmod (M/m_L) + 1) \\
&= -Mx + i_j - 1 + 1 \\
&= -Mx + i_j.
\end{aligned}$$

**Case 2:** $i_j^{(1)}$ even. Then by Lemma 18:

$$h_j^{(1,m_L,d)}\left(\tfrac{(i_j-1)}{M}\right) = -m_L \cdot \frac{(i_j-1)}{M} + i_j^{(1)} = -\frac{m_L \cdot (i_j-1)}{M} + \left\lfloor \frac{m_L \cdot (i_j-1)}{M} \right\rfloor + 1 = 1 - \frac{(i_j-1) \bmod (M/m_L)}{M/m_L}$$

and

$$h_j^{(1,m_L,d)}\left(\tfrac{i_j}{M}\right) = -m_L \cdot \frac{i_j}{M} + i_j^{(1)} = -m_L \cdot \frac{i_j}{M} + \left\lfloor m_L \cdot \frac{(i_j-1)}{M} \right\rfloor + 1 = 1 - \frac{(i_j-1) \bmod (M/m_L)) - 1}{M/m_L}$$

Define $i_j^{(L-1)} := \frac{M}{m_L} - ((i_j - 1) \bmod (M/m_L)).$ Then it holds that $h_j^{(1,m_L,d)}(x) \in \left[\frac{m_L \cdot (i_j^{(L-1)}-1)}{M}, \frac{m_L \cdot (i_j^{(L-1)})}{M}\right]$. Moreover, $i_j^{(L-1)}$ is even if and only if $i_j$ is odd, once more because $\frac{M}{m_L}$ is an even number.

**Case 2.i:** $i_j^{(L-1)}$ is odd (i.e., $i_j$ even). Then follows with the induction hypothesis:

$$
\begin{aligned}
h_j^{(L,\mathbf{m},d)}(x) &= h_j^{(L-1,(m_1,\ldots,m_{L-1}),d)}(h_j^{(1,m_L,d)}(x)) \\
&= \frac{M}{m_L} \cdot (h_j^{(1,m_L,d)}(x)) - (i_j^{(L-1)} - 1) \\
&= \frac{M}{m_L} \cdot (-m_L x + i_j^{(1)}) - (i_j^{(L-1)} - 1) \\
&= -Mx + \frac{M}{m_L} \cdot \left\lfloor \frac{m_L \cdot (i_j - 1)}{M} \right\rfloor + \frac{M}{m_L} - \left( \frac{M}{m_L} - ((i_j - 1) \bmod M/m_L) - 1 \right) \\
&= -Mx + i_j
\end{aligned}
$$

**Case 2.ii:** $i_j^{(L-1)}$ is even (i.e., $i_j$ odd). Then follows with the induction hypothesis:

$$
\begin{aligned}
h_j^{(L,\mathbf{m},d)}(x) &= h_j^{(L-1,(m_1,\ldots,m_{L-1}),d)}(h_j^{(1,m_L,d)}(x)) \\
&= -\frac{M}{m_L} \cdot (h_j^{(1,m_L,d)}(x)) + i_j^{(L-1)} \\
&= -\frac{M}{m_L} \cdot (-m_L x + i_j^{(1)}) + i_j^{(L-1)} \\
&= Mx - \frac{M}{m_L} \cdot \left( \left\lfloor \frac{m^L (i_j - 1)}{M} \right\rfloor + 1 \right) + \frac{M}{m_L} - ((i_j - 1) \bmod (M/m_L) + 1) + 1 \\
&= Mx - (i_j - 1).
\end{aligned}
$$

This concludes the proof for all cases. □

### 5.2.2 PROOF OF LEMMA 5

**Lemma 19.** *Let $d, w \in \mathbb{N}$ and*

$$
R_q = \{x \in \mathbb{R}^d : x_1, \ldots, x_d > 0, \frac{q}{2w} < \|x\|_1 < \frac{q+1}{2w}\}.
$$

*Then* $\operatorname{sgn}(\hat{g}(R_q)) = (-1)^q$ *for all* $q = 0, \ldots, w-1$ *and* $\hat{g}(x) = 0$ *for all* $x \in [0,1]^d$ *with* $\|x\|_1 \geq \frac{1}{2}$.

*Proof.* Let $q \in \{0, \ldots, w-1\}$ and $x \in R_q$. Note that $\hat{g}_0(x) = \mathbf{1}^T x$ for all $q \in \{0, \ldots, w-1\}$.

**Case 1:** $\mathbf{1}^T x < (2q+1)/4w$. This implies $\hat{g}_i(x) = 0 \ \forall q > i$ and $g_i(x) = 2(\mathbf{1}^T x - ((2i-1)/4w)) \ \forall 1 < i \leq q$ and therefore

$$
\hat{g}(x) = \sum_{i=0}^{q} (-1)^i \hat{g}_i(x) = x_1 + \sum_{i=1}^{q} (-1)^i 2(\mathbf{1}^T x - ((2i-1)/4w)).
$$

**Case 1.i:** If $q$ is even, then it holds:

$$
\begin{aligned}
\hat{g}(x) &= \mathbf{1}^T x + \sum_{i=1}^{q/2} 2(\mathbf{1}^T x - ((2(2i)-1)/4w)) - \sum_{i=1}^{q/2} 2(\mathbf{1}^T x - ((2(2i-1)-1)/4w)) \\
&= \mathbf{1}^T x + \sum_{i=1}^{q/2} 2(\mathbf{1}^T x - ((4i-1)/4w)) - \sum_{i=1}^{q/2} 2(\mathbf{1}^T x - ((4i-3)/4w)) \\
&= \mathbf{1}^T x - q/2w > 0
\end{aligned}
$$

**Case 1.ii:** If $q$ is odd, then it holds:

$$\hat{g}(x) = \mathbf{1}^T x + \sum_{i=1}^{(q-1)/2} 2(\mathbf{1}^T x - ((4i-1)/4w)) - \sum_{i=1}^{(q+1)/2} 2(\mathbf{1}^T x - ((4i-3)/4w))$$

$$= \mathbf{1}^T x - 2(q-1)/4w - 2(\mathbf{1}^T x - ((4(q+1)/2) - 3)/4w))$$

$$= -(\mathbf{1}^T x) - 2(q-1)/4w + (4(q+1) - 6)/4w$$

$$= -(\mathbf{1}^T x) + q/2w < 0$$

**Case 2:** $\mathbf{1}^T x \geq (2q+1)/4w$. This implies $\hat{g}_i(x) = 0 \;\; \forall i > q+1$ and $g_i(x) = 2(\mathbf{1}^T x - ((2i-1)/4w)) \;\; \forall 1 < i \leq q+1$ and therefore

$$\hat{g}(x) = \sum_{i=0}^{q+1} (-1)^q \hat{g}_i(x) = x_1 + \sum_{i=1}^{q+1} (-1)^i 2(\mathbf{1}^T x - ((2i-1)/4w)).$$

**Case 2.i:** If $q$ is even, then it holds:

$$\hat{g}(x) = \mathbf{1}^T x + \sum_{i=1}^{q/2} 2(\mathbf{1}^T x - ((4i-1)/4w)) - \sum_{i=1}^{q/2+1} 2(\mathbf{1}^T x - ((4i-3)/4w))$$

$$= \mathbf{1}^T x - 2q/4w - 2(\mathbf{1}^T x - ((4(q/2+1) - 3)/4w))$$

$$= -(\mathbf{1}^T x) - q/w + 2(2q+1)/4w$$

$$= -(\mathbf{1}^T x) + (q+1)/2w > 0$$

**Case 2.ii:** If $q$ is odd, then it holds:

$$\hat{g}(x) = \mathbf{1}^T x + \sum_{i=1}^{(q+1)/2} 2(\mathbf{1}^T x - ((4i-1)/4w)) - \sum_{i=1}^{(q+1)/2} 2(\mathbf{1}^T x - ((4i-3)/4w))$$

$$= \mathbf{1}^T x - (q+1)/2w < 0$$

and hence $\text{sgn}(\hat{g}(x)) = (-1)^q \; \forall x \in R_q \; \forall q = 1, \ldots, w-1$.

Let $x \in [0,1]^d$ with $\mathbf{1}^T x \geq \frac{1}{2}$.

**Case 1:** $w$ even. Then

$$\hat{g}(x) = \sum_{q=0}^{w+1} (-1)^q \cdot \hat{g}_q(x)$$

$$= \mathbf{1}^T x - (\mathbf{1}^T x - \frac{1}{2}) + \sum_{q=1}^{w/2} \hat{g}_{2q}(x) - \hat{g}_{2q-1}(x)$$

$$= \frac{1}{2} + \sum_{q=1}^{w/2} 2(\mathbf{1}^T x - (2 \cdot 2q - 1)/4w) - 2(\mathbf{1}^T x - (2 \cdot (2q-1)) - 1)/4w))$$

$$= \frac{1}{2} + \sum_{q=1}^{w/2} 2(-(4q-1)/4w + (4q-3)/4w)$$

$$= \frac{1}{2} + \sum_{q=1}^{w/1} -1/2w$$

$$= 0$$

**Case 2:** $w$ odd. Then

$$\hat{g}(x) = \sum_{q=0}^{w+1} (-1)^q \cdot \hat{g}_q(x)$$

$$= \hat{g}_0(x) - \hat{g}_w(x) + \hat{g}_{w+1}(x) + \sum_{q=1}^{w-1/2} \hat{g}_{2_q}(x) - \hat{g}_{2_{q-1}}(x)$$

$$= \mathbf{1}^T x - 2(\mathbf{1}^T x - (2w-1)/4w) + \left(\mathbf{1}^T x - \frac{1}{2}\right) + \sum_{q=1}^{w-1/2} -1/w$$

$$= (2w-1)/2w - \frac{1}{2} + \sum_{q=1}^{w-1} -1/2w$$

$$= 1 - 1/2w - \frac{1}{2} - \left(\frac{1}{2} - 1/2w\right)$$

$$= 0$$

$\square$

**Lemma 5.** *Let $d, w \in \mathbb{N}$ and*

$$R_q = \{x \in [0,1]^d : \frac{q}{2w} < \|(1,1,\ldots,1,0) - x\|_1 < \frac{q+1}{2w}\}.$$

*Then there exists a 1-hidden layer neural network $g^{(w,d)} \colon \mathbb{R}^d \to \mathbb{R}$ of width $w+2$ such that*

$$\mathrm{sgn}(g^{(w,d)}(R_q)) = (-1)^q \; \forall q = 0,\ldots,w-1$$

*and $g^{(w,d)}(x) = 0$ for all $x \in [0,1]^d$ with $\|(1,1,\ldots,1,0) - x\|_1 \geq \frac{1}{2}$.*

*Proof.* Let the affine map $t \colon [0,1]^d \to [0,1]^d$ be given by $x \mapsto (1 - x_1, \ldots, 1 - x_{d-1}, x_d)$ and let $\hat{g}$ be the 1-hidden layer neural network from Lemma 19. We prove that the neural network $g := \hat{g} \circ t$ satisfies the assumptions. Let $q \in \{0,\ldots,n-1\}$ and $x \in R_q$. Then $\|(1,1,\ldots,1,0) - t(x)\|_1 = \|(1,1,\ldots,1,0) - (1-x_1,\ldots,1-x_{d-1},x_d)\|_1 = \|x\|_1$. Since $g(x) = g \circ t(x) = \hat{g}(t(x))$, Lemma 19 implies that $\mathrm{sgn}(g(R_q)) = (-1)^q$. Analogously follows that $g(x) = 0$ for all $x \in [0,1]^d$ with $\|(1,1,\ldots,1,0) - x\|_1 \geq \frac{1}{2}$. $\square$

### 5.2.3 Proof of Proposition 6

**Lemma 20.** *Let $g^{(w,d)}$ be the NN from Lemma 5 and $C$ the set of cutting points. Define $R_{q,c} := B^d_{q/(2w \cdot M)}(c) \setminus \overline{B^d_{(q-1)/(2w \cdot M)}(c)}$ for a cutting point $c \in C$ and $q \in \{1,\ldots,w\}$. Then*

1. *$x \in R_{q,c}$ implies $\mathrm{sgn}(g^{(w,d)} \circ h^{(L,\mathbf{m},d)}(x)) = (-1)^q$ and*

2. *$x \notin \bigcup\limits_{q \in [w], c \in C} R_{q,c}$ implies $\mathrm{sgn}(g^{(w,d)} \circ h^{(L,\mathbf{m},d)}(x)) = 0$.*

*In particular, $g^{(w,d)} \circ h^{(L,\mathbf{m},d)}(x) = 0$ for all $x \in \partial R_{q,c}$.*

In order to count the annuli we need to count the cutting points.

*Proof.* By definition of $c$ being a cutting point, there exist odd numbers $i_1,\ldots,i_{d-1} \in [M]$ and an even number $i_d \in [M]$ such that $c = (\frac{i_1-1}{M},\ldots,\frac{i_d-1}{M})$. Let $x \in [0,1]^d$ with $\|x - c\|_\infty \leq \frac{1}{M}$, then either $x_j \in \left[\frac{i_j-2}{M}, \frac{i_j-1}{M}\right]$ or $x_j \in \left[\frac{i_j-1}{M}, \frac{i_j}{M}\right]$. Let $J^+ := \{j \in [d] : x_j - c_j \leq 0\}$ be the set of

indices $j$ such that $x_j \in \left[\frac{i_j - 1}{M}, \frac{i_j}{M}\right]$ and $J^- := [d] \setminus J^+$. Let $y = h^{(L,\mathbf{m},d)}(x) \in [0,1]^d$. Then it follows with Lemma 4 that

$$
\begin{aligned}
\mathbf{1}^T y &= \sum_{j \in J^+} M \cdot x_j - (i_j - 1) + \sum_{j \in J^-} -M \cdot x_j + (i_j - 1) \\
&= \sum_{j \in J^+} M \cdot (x_j - c_j + c_j) - (i_j - 1) + \sum_{j \in J^-} -M \cdot (x_j - c_j + c_j) + (i_j - 1) \\
&= \sum_{j \in J^+} M \cdot (x_j - c_j) + \sum_{j \in J^+} M \cdot c_j - (i_j - 1) \\
&\quad + \sum_{j \in J^-} -M \cdot (x_j - c_j) + \sum_{j \in J^-} -M \cdot c_j + (i_j - 1) \\
&= \sum_{j \in J^+} M \cdot (x_j - c_j) + \sum_{j \in J^-} -M \cdot (x_j - c_j) \\
&= M \cdot \sum_{j=1}^d |x_j - c_j| \\
&= M \cdot \|x - c\|_1
\end{aligned}
$$

If $x \in R_{q,c}$, then in particular $\|x - c\|_\infty \leq \frac{1}{M}$ and thus:

$$
\mathbf{1}^T y = M \cdot \|x - c\|_1 < M \cdot \frac{q+1}{M \cdot 2n} = \frac{q+1}{2n}
$$

and

$$
\mathbf{1}^T y = M \cdot \|x - c\|_1 > M \cdot \frac{q}{M \cdot 2n} = \frac{q}{2n}.
$$

With Lemma 5 it follows that $\mathrm{sgn}(g(y)) = (-1)^q$ and therefore $g \circ h^{(L,\mathbf{m},d)}(x) = g(y)$ concludes the first case.

If $x$ is not in any $R_{q,c}$, then either $x \in \partial R_{q,c}$ for some cutting point $c$ or it holds that $\|x - c\|_1 \geq \frac{1}{2 \cdot M}$ for every cutting point $c$. In the first case it follows directly from the above shown that $g \circ h^{(L,\mathbf{m},d)}(x)) = 0$, since $g \circ h^{(L,\mathbf{m},d)}$ is continuous. In the second case there exists a cutting point $c$ such that $\|x - c\|_\infty \leq \frac{1}{M}$, since for every $x_j$ either $\lfloor M \cdot x_j \rfloor$ or $\lceil M \cdot x_j \rceil$ is even. Thus $\mathbf{1}^T h^{(L,\mathbf{m},d)}(x) = M \cdot \|x - c\|_1 \geq M \cdot \frac{1}{2 \cdot M} = \frac{1}{2}$ and therefore it follows with Lemma 5 that $g \circ h^{(L,\mathbf{m},d)}(x) = 0$, which concludes the proof.

$\square$

**Observation 21.** *Cutting points lie on a grid in the unit cube, with $\frac{M}{2}$ many cutting points into dimensions $1, \ldots, d-1$ and $\frac{M}{2} + 1$ many in dimension $d$. Thus, there are $\frac{M^{(d-1)}}{2^{d-1}} \cdot \left(\frac{M}{2} + 1\right)$ cutting points. Note that since $M$ is an even number, these points cannot lie on the boundary unless the last coordinate is $0$ or $M$. This means, $2 \cdot \frac{M^{(d-1)}}{2^{d-1}} = \frac{M^{(d-1)}}{2^{d-2}}$ of the cutting points are located on the boundary of the unit cube and the remaining $\frac{M^{(d-1)}}{2^{d-1}} \cdot \left(\frac{M}{2} - 1\right)$ are in the interior.*

**Proposition 6.** *The space $Y_{d,w}$ is homeomorphic to the disjoint union of $p_d = \frac{M^{(d-1)}}{2^{d-1}} \cdot \left(\frac{M}{2} - 1\right) \cdot \left\lceil \frac{w}{2} \right\rceil$ many $(d-1)$-annuli and $p'_d = \frac{M^{(d-1)}}{2^{d-2}} \cdot \left\lceil \frac{w}{2} \right\rceil$ many disks, that is,*

$$
Y_{d,w} \cong \coprod_{k=1}^{p_d} (S^{d-1} \times [0,1]) \sqcup \coprod_{k=1}^{p'_d} D^d.
$$

*Proof.* We observe that the sets $Y_{d,w} \cap B^d_{1/2M}(x)$ are disjoint for cutting points $x$ because we have $\|x - x'\|_1 \geq \frac{2}{M}$ for any two distinct cutting points $x, x'$. Moreover, by Lemma 20, we have $(g \circ h)(y) = 0$ for all $y \in \partial B^d_{1/2M}(x)$ for $x \in C$. Therefore, the sets $Y_{d,w} \cap \overline{B}^d_{1/2M}(x)$ are pairwise disjoint for $x \in C$. Since

$$
\coprod_{x \in E} Y_{d,w} \cap B^d_{1/2M}(x) = Y_{d,w},
$$

the number of cutting points of the interior is $\frac{M \cdot (d-1)}{2^{d-1}} \cdot \left(\frac{M}{2} - 1\right) \cdot \left\lceil \frac{n}{2} \right\rceil$ and the number of the cutting points on the boundary is $\frac{M \cdot (d-1)}{2^{d-2}} \cdot \left\lceil \frac{n}{2} \right\rceil$ by Observation 21, it suffices to show that $Y_{d,w} \cap \overline{B}^d_{1/2M}(x) \cong \coprod_{i=1}^{\left\lceil \frac{n}{2} \right\rceil} S^{d-1} \times D^1$ for every $x \in C \cap \mathrm{int}([0,1]^d)$ and $Y_{d,w} \cap \overline{B}^2_{1/2M}(x) \cong \coprod_{i=1}^{\left\lceil \frac{n}{2} \right\rceil} D^d$ for every $x \in C \cap \partial[0,1]^d$.

By Lemma 20, we can see that for every $x \in C \cap \mathrm{int}([0,1]^d)$, we have

$$
\begin{aligned}
Y_{d,w} \cap \overline{B}^d_{1/2M}(x) &= \coprod_{1 \leq q \leq w \text{ odd}} B^d_{q/(w \cdot 2M)}(x) \setminus \overline{B^d_{(q-1)/(w \cdot 2M)}(x)} \\
&\cong \coprod_{1 \leq q \leq w \text{ odd}} S^{d-1} \times [0,1] \\
&= \coprod_{q=1}^{\left\lceil \frac{w}{2} \right\rceil} S^{d-1} \times [0,1],
\end{aligned}
$$

as well as for every $x \in C \cap \partial([0,1]^d)$, we have

$$
Y_{d,w} \cap \overline{B}^d_{1/2M}(x) \cong \coprod_{1 \leq q \leq w \text{ odd}} \left( B^d_{q/(w \cdot 2M)}(x) \setminus \overline{B^d_{(q-1)/(w \cdot 2M)}(x)} \right) \cap [0,1]^d \cong \coprod_{q=1}^{\left\lceil \frac{w}{2} \right\rceil} D^d,
$$

proving the claim. $\qquad \square$

### 5.2.4 PROOF OF LEMMA 7

**Lemma 7.** *For $k \leq d$ and $\mathbf{w} = (w_1, \ldots, w_{d-1}) \in \mathbb{N}^{d-1}$ it holds that*

1. *$f^{(w_1,\ldots,w_{k-2})} \circ p_{k-1}(x) \neq 0 \implies g^{(w_{k-1},k)}(x) = 0$ and*

2. *$g^{(w_{k-1},k)}(x) \neq 0 \implies f^{(w_1,\ldots,w_{k-2})} \circ p_{k-1}(x) = 0$*

*for all $x \in [0,1]^k$.*

*Proof.* We adopt the notation $c^{(k)} := (1,1,\ldots,1,0) \in \mathbb{R}^k$ throughout.

We first show that for all $x \in [0,1]^k$,

$$
\|x - c^{(k)}\|_1 \leq \frac{1}{2} \Rightarrow g^{(w_{k-2},k-1)} \circ p_{k-1}(x) = 0. \tag{1}
$$

Let $x \in [0,1]^k$ with $g^{(w_{k-2},k-1)} \circ p_{k-1}(x) \neq 0$. Lemma 5 implies that $\|p_{k-1}(x) - c^{(k-1)}\|_1 < \frac{1}{2}$. Therefore we have $\frac{1}{2} > |\pi_{k-1} \circ p_{k-1}(x) - 0| = |\pi_{k-1}(x) - 0| = x_{k-1}$ which also means $|x_{k-1} - 1| > \frac{1}{2}$ and thus $\|x - c^{(k)}\|_1 > \frac{1}{2}$.

Note that by Lemma 5 it suffices to show that $f^{(w_1,\ldots,w_{k-1})} \circ p_{k-1}(x) = 0$ for all $x$ with $\|x - c^{(k)}\|_1 \leq \frac{1}{2}$. We prove this by induction over $k$. The base case has already been covered since $g^{(w_1,2)} = f^{w_1}$. Furthermore

$$
\begin{aligned}
f^{(w_1,\ldots,w_{k-2})} \circ p_{k-1} &= (f^{(w_1,\ldots,w_{k-3})} \circ p_{k-2} + g^{(w_{k-2},k-1)}) \circ p_{k-1} \\
&= f^{(w_1,\ldots,w_{k-3})} \circ p_{k-2} \circ p_{k-1} + g^{(w_{k-2},k-1)} \circ p_{k-1} \\
&= f^{(w_1,\ldots,w_{k-3})} \circ p_{k-2} + g^{(w_{k-2},k-1)} \circ p_{k-1}
\end{aligned}
$$

and thus the induction hypothesis and (1) imply that $f^{\mathbf{w}} \circ p_{k-1}(x) = 0$ for $x$ with $\|x - c^{(d)}\|_1 \leq \frac{1}{2}$.
$\qquad \square$

### 5.2.5 PROOF OF LEMMA 8

**Lemma 22.** *The following diagram commutes:*

$$
\begin{array}{ccc}
[0,1]^k & \xrightarrow{\;h^{(L,\mathbf{m},k)}\;} & [0,1]^k \\
\downarrow{\scriptstyle p_{k-1}} & & \downarrow{\scriptstyle p_{k-1}} \quad \searrow{\scriptstyle f^{\mathbf{w}}\circ p_{k-1}} \\
[0,1]^{k-1} & \xrightarrow[\;h^{(L,\mathbf{m},k-1)}\;]{} & [0,1]^{k-1} \xrightarrow[\;f^{\mathbf{w}}\;]{} \mathbb{R}
\end{array}
$$

*Proof.* In order to show that the left half of the diagram commutes, we prove that

$$(\pi_j \circ h^{(L,\mathbf{m},k-1)} \circ p_{k-1})(x) = (\pi_j \circ p_{k-1} \circ h^{(L,\mathbf{m},k)})(x)$$

for every $j \in \{1, \dots, k-1\}$ and $x \in [0,1]^k$. For any $x \in [0,1]^k$, there exist indices $i_1, \dots, i_k$ such that $x = (x_1, \dots, x_k) \in W^{(L,\mathbf{m},k)}_{(i_1,\dots,i_k)}$. Moreover, if $x \in W^{(L,\mathbf{m},k)}_{(i_1,\dots,i_k)}$, we have $p_{k-1}(x) \in W^{(L,\mathbf{m},k-1)}_{(i_1,\dots,i_{k-1})}$ because

$$p_{k-1}\left(W^{(L,\mathbf{m},k)}_{(i_1,\dots,i_k)}\right) = p_{k-1}\left(\prod_{j=1}^{k}\left[\frac{(i_j-1)}{M}, \frac{i_j}{M}\right]\right) = \prod_{j=1}^{k-1}\left[\frac{(i_j-1)}{M}, \frac{i_j}{M}\right].$$

We use this observation combined with Lemma 4, assuming that $i_j$ is odd:

$$
\begin{aligned}
(\pi_j \circ h^{(L,\mathbf{m},k-1)} \circ p_{k-1})(x) &= M \cdot (p_{k-1}(x))_j - (i_j - 1) \\
&= M \cdot x_j - (i_j - 1) \\
&= (\pi_j \circ h^{(L,\mathbf{m},k)})(x) \\
&= (\pi_j \circ p_{k-1} \circ h^{(L,\mathbf{m},k)})(x),
\end{aligned}
$$

as claimed. The case where $i_j$ is even follows analogously. $\qquad\square$

**Lemma 8.** *For $2 \leq k \leq d$, the space $X_k := (f^{(w_1,\dots,w_{k-1})} \circ h^{(L,\mathbf{m},k)})^{-1}((-\infty,0))$ satisfies*

$$X_k = (X_{k-1} \times [0,1]) \sqcup Y_{k,w}$$

*with $X_1 := \emptyset$.*

*Proof.* For $k = 2$ it holds that $f^{w_1} = g^{(w_1,2)}$ and therefore the claim holds trivially. Now let $k \geq 3$. Since $f^{(w_1,\dots,w_{k-1})} = f^{(w_1,\dots,w_{k-2})} \circ p_{k-1} + g^{(w_{k-1},k)}$ and the spaces $(f^{(w_1,\dots,w_{k-2})} \circ p_{k-1} \circ h^{(L,\mathbf{m},k)})^{-1}((-\infty,0))$ and $(g^{(w_{k-1},k)} \circ h^{(L,\mathbf{m},k)})^{-1}((-\infty,0))$ are disjoint by Lemma 7, it follows that

$$
\begin{aligned}
&(f^{(w_1,\dots,w_{k-1})} \circ h^{(L,\mathbf{m},k)})^{-1}((-\infty,0)) \\
&= ((f^{(w_1,\dots,w_{k-2})} \circ p_{k-1} + g^{(w_{k-1},k)}) \circ h^{(L,\mathbf{m},k)})^{-1}((-\infty,0)) \\
&= (f^{(w_1,\dots,w_{k-2})} \circ p_{k-1} \circ h^{(L,\mathbf{m},k)} + g^{(w_{k-1},k)} \circ h^{(L,\mathbf{m},k)})^{-1}((-\infty,0)) \\
&= (f^{(w_1,\dots,w_{k-2})} \circ p_{k-1} \circ h^{(L,\mathbf{m},k)})^{-1}((-\infty,0)) \sqcup (g^{(w_{k-1},k)} \circ h^{(L,\mathbf{m},k)})^{-1}((-\infty,0)) \\
&= (f^{(w_1,\dots,w_{k-2})} \circ h^{(L,\mathbf{m},k-1)} \circ p_{k-1})^{-1}((-\infty,0)) \sqcup (g^{(w_{k-1},k)} \circ h^{(L,\mathbf{m},k)})^{-1}((-\infty,0)) \\
&= X_{k-1} \times [0,1] \sqcup Y_{k,w},
\end{aligned}
$$

where the second last equality is due to Lemma 22. $\qquad\square$

### 5.2.6 PROOF OF THEOREM 24

**Lemma 23.** *The space $Y_{d,w} := (g^{(w,d)} \circ h^{(L,\mathbf{m},d)})^{-1}((-\infty,0))$ satisfies*

(i) $H_0(Y_{d,w}) \cong \mathbb{Z}^{p+p'}$,

(ii) $H_{d-1}(Y_{d,w}) \cong \mathbb{Z}^p$,

(iii) $H_k(Y_{d,w}) = 0$ for $k \geq d$

with $p = \frac{M^{(d-1)}}{2^{d-1}} \cdot \left(\frac{M}{2} - 1\right) \cdot \left\lceil \frac{n}{2} \right\rceil$ and $p' = \frac{M^{(d-1)}}{2^{d-2}} \cdot \left\lceil \frac{n}{2} \right\rceil$

*Proof.* Follows directly from Observation 16 and Proposition 6 and the disjoint union axiom (Proposition 15). $\square$

**Theorem 24.** *Let $d \in \mathbb{N}$, then there is a ReLU NN $F \colon \mathbb{R}^d \mapsto \mathbb{R}$ with architecture $(d, m_1 \cdot d, \dots, m_L \cdot d, n_{L+1}, 1)$ such that the space $X_d := F^{-1}((-\infty, 0))$ satisfies*

(i) $\beta_0(X_d) = \sum_{k=1}^{d-1} \frac{M^k}{2^k} \cdot \left(\frac{M}{2} + 1\right) \cdot \left\lceil \frac{w_k}{2} \right\rceil$

(ii) $\beta_k(X_d) = \frac{M^{(k-1)}}{2^{k-1}} \cdot \left(\frac{M}{2} - 1\right) \cdot \left\lceil \frac{w_{k-1}}{2} \right\rceil$ *for $0 < k < d$.*

*Proof.* We consider the map $F := f^{\mathbf{w}} \circ h^{(L,\mathbf{m},d)}$ that was previously constructed (Lemma 8). For $d = 2$, the statement is identical to Lemma 23. Indeed, we have

$$2 \cdot \frac{\left(\frac{M}{2} + 1\right)^3 - 1}{M} - \frac{M}{2} - 2 = \left(\frac{M}{2} + 1\right)^2 + \frac{M}{2} + 1 - \frac{M}{2} - 2$$
$$= \frac{M}{2}\left(\frac{M}{2} + 1\right).$$

Let $d \geq 3$. Using Proposition 6, we see that

$$H_k(X_d) \cong H_k(X_{d-1} \sqcup Y_{d,w}) \cong H_k(X_{d-1}) \oplus \prod_{i=1}^{p_d} H_k(S^{d-1}) \oplus \prod_{i=1}^{p'_d} H_k(D^d) \tag{2}$$

and therefore

$$\beta_k(X_d) = \beta_k(X_{d-1}) + \sum_{i=1}^{p_d} \left(\beta_k(S^{d-1})\right) + \sum_{i=1}^{p'_d} \beta_k(D^d) \tag{3}$$

where $p_d = \frac{M^{d-1}}{2^{d-1}} \cdot \left(\frac{M}{2} - 1\right) \cdot \left\lceil \frac{w_{d-1}}{2} \right\rceil$ and $p'_d = \frac{M^{d-1}}{2^{d-2}} \cdot \left\lceil \frac{w_{d-1}}{2} \right\rceil$. Fix some $k \in \mathbb{N}$. For different values of $k$, we obtain the claims:

(i) For $k = 0$, equation (3) implies

$$\beta_0(X_d) = \beta_0(X_{d-1}) + p_d + p'_d = \sum_{i=2}^{d} \frac{M^{(i-1)}}{2^{i-1}} \cdot \left(\frac{M}{2} + 1\right) \cdot \left\lceil \frac{w_{i-1}}{2} \right\rceil$$

(ii) For $k \leq d - 1$, we have $\beta_{d-1}(X_d) = 0$ and therefore

$$\beta_{d-1}(X_d) = p_d = \left(\frac{M}{2} - 1\right) \cdot \frac{M^{d-1}}{2^{d-1}} \cdot \left\lceil \frac{w_{d-1}}{2} \right\rceil.$$

For $0 < k < d - 1$, we have $\beta_k(X_d) = \beta_k(X_{d-1})$, i.e., the claim is satisfied by induction.

(iii) Finally for $k \geq d$, we observe that all summands of (3) vanish.

$\square$

### 5.2.7 PROOF OF COROLLARY 11

**Corollary 11.** *Let $A$ be the architecture as in Theorem 24, then there is a ReLU NN $F\colon \mathbb{R}^d \mapsto \mathbb{R}$ with architecture $A$ such that the space $X_d := F^{-1}((-\infty, 0))$ satisfies $\chi(X_d) \in \Omega\left(M^d \cdot \sum_{i=1}^{d-1} w_i\right)$, where $\chi(X_d)$ denotes the Euler characteristic of the space $X_d$.*

*Proof.* The Euler characteristic of a finite CW complex $X$ is given by the alternating sum of its Betti numbers, i.e., by the sum $\sum_{k\in\mathbb{N}}(-1)^k\beta_k(X)$. By Theorem 24, this term is dominated by the zeroth Betti number, from which the claim follows. $\square$

The Euler characteristic is an invariant used widely in differential geometry in addition to algebraic topology. For instance, it can also be defined by means of the index of a vector field on a compact smooth manifold.

### 5.3 STABILITY

Before we prove the stability of our construction, we prove stability for a wider range of neural networks. Througout this section we will use definitions and statements from Section 5.1.1.

**Definition 11** (The realization map). *Let $(n_0, \ldots n_{L+1})$ be an architecture, $K \subseteq \mathbb{R}^d$ a polyhedron, $\mathbb{R}^D \cong \bigoplus_{\ell=1}^{L+1} \mathbb{R}^{(n_{\ell-1}+1)\times n_\ell}$ the corresponding parameter space where the vector space isomorphism is given by $p \mapsto (A^{(\ell)}(p), b^{(\ell)}(p))_{\ell=1,\ldots,L+1}$ for $A^{(\ell)}(p) \in \mathbb{R}^{n_{\ell-1}\times n_\ell}, b^{(\ell)}(p) \in \mathbb{R}^{n_\ell}$ and $\ell = 1, \ldots, L + 1$. We define $\Phi\colon \mathbb{R}^D \to C(K)$ to be the realization map, that assigns to a vector of weights the function the corresponding neural network computes, i.e.,*

$$\Phi(p) := T_{L+1}(p) \circ \sigma_{n_L} \circ T_L(p) \circ \cdots \circ \sigma_{n_1} \circ T_1(p)$$

*where $T_\ell(p)\colon \mathbb{R}^{n_{\ell-1}} \to \mathbb{R}^{n_\ell}, x \mapsto A^{(\ell)}(p)x + b^{(\ell)}(p)$*

*Furthermore let*

$$\Phi^{(\ell)}(p) := T_\ell(p) \circ \sigma_{n_\ell} \circ \cdots \circ \sigma_{n_1} \circ T_0(p)$$

*and*

$$\Phi^{(i,\ell)}(p) := \pi_i \circ T_\ell(p) \circ \cdots \circ \sigma_{n_1} \circ T_0(p).$$

*We denote the points of non-linearity introduced by the $i$-th neuron in the $\ell$-th layer by*

$$\tilde{H}_{i,\ell}(p) := H\left(A_i^{(\ell)}(p), b_i^{(\ell)}(p)\right)$$

**Definition 12** (Canonical polyhedral complex (Grigsby et al. (2022))). *Let $p$ be a vector of weights. Recall that $\Phi(p)$ is the corresponding neural network.*

*We iteratively define polyhedral complexes $\mathcal{P}^{(\ell,i)}(p)$ by $\mathcal{P}^{(1,0)}(p) := \{K\}$ and*

$$\mathcal{P}^{(\ell,i)}(p) := \{R \cap (\Phi^{(\ell-1)}(p))^{-1}(\tilde{H}_{i,\ell}^s(p)) \mid R \in \mathcal{P}^{(\ell,i-1)}(p), s \in \{-1,0,1\}\}$$

*for $i = 2, \ldots n_\ell, \ell = 1, \ldots L$ and*

$$\mathcal{P}^{(\ell,1)}(p) := \{R \cap (\Phi^{(\ell-1)}(p))^{-1}(\tilde{H}_{i,\ell}^s(p)) \mid R \in \mathcal{P}^{(\ell-1,n_{\ell-1})}(p), s \in \{-1,0,1\}\}$$

*for $\ell = 1, \ldots L$.*

Note that for all $j \leq i$, it holds that $\Phi^{(j,\ell)}(p)$ is affine linear on $R$ for each $R \in \mathcal{P}^{(\ell,i)}(p)$ and we denote this affine linear map by $\Phi_{|R}^{(j,\ell)}(p)$. For $\ell \in [L], i \in [n_\ell]$ and $R \in \mathcal{P}_d^{(i,\ell)}(p)$ we denote the points of non-linearity in the region $R$ introduced by the $i$-th neuron in the $\ell$-th layer with respect to the first $\ell - 1$ layer map by

$$H_{i,\ell,R}(p) := (\Phi_{|R}^{(\ell-1)}(p))^{-1}(\tilde{H}_{i,\ell}(p)) = H\left(A_i^{(\ell)}(p)\left(\Phi_{|R}^{(\ell-1)}(p)(x)\right), b_i^{(\ell)}(p)\right).$$

For the sake of simplification we set $\mathcal{P}^{(0,\ell)} := \mathcal{P}^{(n_{\ell-1},\ell-1)}$. Furthermore, since for $R \in \mathcal{P}_d^{(\ell,i-1)}(p), F \in \mathcal{P}^{(\ell,i-1)}(p), F \preceq R$ it holds that $F \cap H_{i,\ell,R}^s(p) = F \cap (\Phi^{(\ell-1)}(p))^{-1}(\tilde{H}_{i,\ell}^s(p))$ due to the continuity of the function $\Phi(u)$, we have that

$$\mathcal{P}^{(\ell,i)}(p) = \{F \cap H_{i,\ell,R}^s(p) \mid R \in \mathcal{P}_d^{(\ell,i-1)}(p), F \in \mathcal{P}^{(\ell,i-1)}(p), F \preceq R, s \in \{-1,0,1\}\}$$

We call $\mathcal{P}(u) := \mathcal{P}^{(n_L,L)}$ the *canonical polyhedral complex* of $\Phi(u)$.

**Definition 13.** *Let $K$ be a polytope and $\Phi(p)\colon K \to \mathbb{R}$ be a ReLU neural network of architecture $(n_0, \ldots n_{L+1})$. Then we call $\Phi(p)$ stable if for every $\ell \in [L+1], i \in [n_\ell]$ and all $R \in \mathcal{P}_d^{(i-1,\ell)}(p)$ it holds*

1. $\dim(H_{i,\ell,R}(p)) = d - 1$ *and*

2. $H_{i,\ell,R}(p) \cap R_0 = \emptyset$.

**Proposition 25.** *Let $K$ be a polytope and $\Phi(p)\colon K \to \mathbb{R}$ be a stable ReLU neural network of architecture $(n_0, \ldots n_{L+1})$. Then for every $\varepsilon > 0$, there is a an open set $U \subseteq \mathbb{R}^D$ such that for every $u \in U$ there is an $\varepsilon$-isomorphism $\varphi_u \colon \mathcal{P}(p) \to \mathcal{P}(u)$.*

*Proof.* We will prove the following stronger statement by induction on the indexing of the neurons.

**Claim.** *For every $\ell \in [L+1]$, $i \in [n_\ell]$ and every $\varepsilon > 0$, there is a $\delta > 0$ such that for all $u \in B_\delta^{\|\cdot\|_\infty}(p)$ there is an $\varepsilon$-isomorphism $\varphi_u^{(i,\ell)} \colon \mathcal{P}^{(i,\ell)}(p) \to \mathcal{P}^{(i,\ell)}(u)$.*

The induction base is trivially satisfied.
So we assume that the statement holds for $(i-1, \ell)$. For simpler notation we denote $\varphi_u^{(i-1,\ell)}$ by $\varphi_u$ and $H_{i,\ell,R}(p)$ by $H_R(p)$. Let $\varepsilon > 0$ and $F \in \mathcal{P}^{(i-1,\ell)}(p)$. There is an $R \in \mathcal{P}_d^{(i-1,\ell)}(p)$ such that $F \preceq R$. In the following we wish to find a $\delta_F > 0$ such that there are $\varepsilon$-isomorphisms

$$\varphi_{(u,R,s)}^{(i,\ell)} \colon F \cap H_R^s(p) \to \varphi_u(F) \cap H_{\varphi_u(R)}^s(u)$$

for $s \in \{-1, 0, 1\}$ and all $u \in B_{\delta_F}^{\|\cdot\|_\infty}(p)$.

Since $\Phi(p)$ is stable, we obtain by Lemma 14 a $\delta_2 > 0$ such that for all $\delta_2$-isomorphisms $\varphi \colon F \to Q$ there are $\frac{\varepsilon}{3}$-isomorphisms $\gamma^s \colon F \cap H_R^s(p) \to \varphi(F) \cap H_R^s(p)$. By the induction hypothesis we obtain $\delta_1 > 0$ such that for all $u \in B_{\delta_1}^{\|\cdot\|_\infty}(p)$ there is an $\delta_2$-isomorphism $\varphi_u \colon \mathcal{P}(p)^{(i-1,\ell)} \to \mathcal{P}(u)^{(i-1,\ell)}$ and hence we obtain $\frac{\varepsilon}{3}$-isomormorphisms

$$\gamma^{(s,F)} \colon F \cap H_R^s(p) \to \varphi_u(F) \cap H_R^s(p).$$

Let $H_{\varphi_u(R)}(u, p) := H_R(u_{1,1}, \ldots u_{i-1,\ell}, p_{i,\ell}, \ldots p_{n_{L+1},L+1})$ with $u_{j,k}, p_{j,k} \in \mathbb{R}^{n_k}$ being the parameters associated to the $j$-th neuron in the $k$-th layer. Again, for simpler notation, let the affine maps $\Phi_{|R}^{(\ell-1)}(p)$ be given by $x \mapsto Mx + c$ and $\Phi_{|\varphi_u(R)}^{(\ell-1)}(u)$ by $x \mapsto Nx + d$ and the non-linearity points introduced by the $i$-th neuron in the $\ell$-th layer by $\tilde{H}_{i,\ell}(p) = H(a,b)$. Then we have that

$$H_R(p) = H(a^T M, a^T c + b)$$

and

$$H_{\varphi_u(R)}(u, p) = H(a^T N, a^T d + b).$$

By Lemma 14 we know that $(\varphi_u(F))_0 \cap H_R(p) = \emptyset$ and hence by Lemma 13 there is a $\delta_3 > 0$ such that there are $\frac{\varepsilon}{3}$-isomorphisms $\psi^s \colon \varphi_u(F) \cap H_R^s(p) \to \varphi_u(F) \cap H^s(y,z)$ for all $(y,z) \in B_{\delta_3}^{d+1}((a^T M, a^T c + b))$. Let $C := n_{\ell-1} \max\limits_{i=1,\ldots,n_{\ell-1}} \{a_i\}$ and $u \in \mathbb{R}^D$ with $\|u - p\|_\infty < \frac{\delta_3}{C}$. Then we have that

$$\|(a^T M, a^T c + b) - (a^T N, a^T d + b)\|_\infty = \max_{i=1,\ldots,d} \left\{ \sum_{j=1}^{n_{\ell-1}} a_j(m_{ij} - n_{ij}), \sum_{j=1}^{n_{\ell-1}} a_j(c_j - d_j) \right\}$$

$$< \max_{i=1,\ldots,d} \left\{ \sum_{j=1}^{n_{\ell-1}} a_j \frac{\delta_3}{C}, \sum_{j=1}^{n_{\ell-1}} a_j \frac{\delta_3}{C} \right\}$$

$$< \delta_3$$

and hence there are $\frac{\varepsilon}{3}$-isomorphisms

$$\psi^{(s,F)} \colon \varphi_u(F) \cap H_R^s(p) \to \varphi_u(F) \cap H_{\varphi_u(R)}^s(u, p)$$

By Lemma 13 we know that $(\varphi_u(F))_0 \cap H_{\varphi(R)}(u,p) = \emptyset$ and hence by the same lemma there is a $\delta_4 > 0$ such that there are $\frac{\varepsilon}{3}$-isomorphisms $\alpha^s \colon \varphi_u(F) \cap H^s_{\varphi(R)}(u,p) \to \varphi_u(F) \cap H^s(y,z)$ for all $(y,z) \in B^{d+1}_{\delta_4}((a^T N, a^T d + b))$. Let $a' \in \mathbb{R}^{n_{\ell-1}}, b' \in \mathbb{R}$ such that $\tilde{H}_{i,\ell}(u) = H(a'^T, b')$. Then we have that

$$H_{\varphi_u(R)}(u) = H(a'^T N, a'^T d + b').$$

Let $E := n_{\ell-1} \max\limits_{i,j=1,\ldots,n_{\ell-1}} \{n_{ij}, d_j\}$ and $u \in \mathbb{R}^D$ with $\|u - p\|_\infty < \frac{\delta_5}{E}$. Then we have that

$$\|(a'^T N, a'^T d + b') - (a^T N, a^T d + b)\|_\infty = \max\limits_{i=1,\ldots,d} \left\{ \sum_{j=1}^{n_{\ell-1}} n_{ij}(a'_j - a_j), \left( \sum_{j=1}^{n_{\ell-1}} d_j(a'_j - a_j) \right) + (b'_j - b_j) \right\}$$

$$< \max\limits_{i=1,\ldots,d} \left\{ \sum_{j=1}^{n_{\ell-1}} n_{ij}\frac{\delta_5}{E}, \sum_{j=1}^{n_{\ell-1}} n_{ij}\frac{\delta_5}{E} \right\}$$

$$< \delta_4$$

and hence there are $\frac{\varepsilon}{3}$-isomorphisms

$$\alpha^{(s,F)} \colon \varphi_u(F) \cap H^s_{\varphi(R)}(u,p) \to \varphi_u(F) \cap H^s_{\varphi_u(R)}(u).$$

Let $\delta_F := \min\{\delta_2, \frac{\delta_4}{C}, \frac{\delta_5}{E}\}$, then for all $u \in B^{D,\|\cdot\|_\infty}_{\delta_F}(p)$ there is an $\varepsilon$-isomorphism

$$\varphi^{(i,\ell)}_{(u,F,s)} \colon F \cap H^s_R(p) \to \varphi_u(F) \cap H^s_{\varphi_u(R)}(u)$$

given by

$$\varphi^{(i,\ell)}_{(u,F,s)} = \alpha^{(s,F)} \circ \psi^{(s,F)} \circ \gamma^{(s,F)}.$$

Lastly, let $\delta = \min\{\delta_F \mid F \in \mathcal{P}^{(i-1,\ell)}(p)\}$. Since every element of $\mathcal{P}^{(i,\ell)}(p)$ is of the form $F \cap H^s_R(p)$, it now remains to show that the map $\varphi^{(i,\ell)}_u \colon \mathcal{P}^{(i,\ell)}(p) \to \mathcal{P}^{(i,\ell)}(u)$ defined by

$$\varphi^{(i,\ell)}_u(F \cap H^s_R(p)) := \varphi_u(F) \cap H^s_{\varphi_u(R)}(u)$$

is an $\varepsilon$-isomorphism for all $u \in B^{\|\cdot\|_\infty}_\delta(p)$. Since $\varphi_u$ and $\varphi^{(i,\ell)}_{(u,F,s)}$ are bijections, the same holds for $\varphi^{(i,\ell)}_u$. Furthermore let $G \preceq F \cap H^s_R(p)$, then there is a $G' \preceq F$ and a $s' \in \{0, s\}$ such that $G = G' \cap H^{s'}_R(p)$. Since $\varphi_u$ is an isomorphism by the induction hypothesis, it follows that

$$\varphi^{(i,\ell)}_u(G' \cap H^{s'}_R(p)) = \varphi_u(G') \cap H^{s'}_{\varphi_u(R)}(u) \preceq \varphi_u(F) \cap H^s_{\varphi_u(R)}(u)$$

and hence $\varphi^{(i,\ell)}_u$ is an $\varepsilon$-isomorphism as claimed. □

**Definition 14.** *Let $K$ be a polytope and $\Phi(p) \colon K \to \mathbb{R}$ be a ReLU neural network of architecture $(n_0, \ldots, n_L, 1)$. Then we call $\Phi(p)$ topologically stable if for all $R \in \mathcal{P}^{(n_L,L)}_d(p)$ it holds that*

1. *$\dim(H_{1,L+1,R}(p)) = d - 1$ and*

2. *$H_{1,L+1,R}(p) \cap R_0 = \emptyset$.*

**Proposition 26.** *Let $\Phi(p)$ be a topologically stable ReLU neural network, then there is a $\delta > 0$, such that for all $u \in B_\delta(p)$ it holds that $K \cap \Phi(p)^{-1}((-\infty, 0))$ is homeomorphic to $K \cap \Phi(u)^{-1}((-\infty, 0))$*

*Proof.* Let $\mathcal{P}^-(p) := \{F \cap H^s_{1,L+1,R}(p) \mid R \in \mathcal{P}_d(p), F \in \mathcal{P}(p), F \preceq R, s \in \{-1, 0\}\}$ be the polyhedral complex consisting of all subpolyhedron of $\mathcal{P}(p)$ where $\Phi(p)$ takes on non-negative values. Analogously to the proof of Proposition 25 we obtain a $\delta > 0$ such that $\mathcal{P}^-(p)$ and $\mathcal{P}^-(u)$ are isomorphic as polyhedral complexes and hence in particular there is a homeomorphism $\varphi \colon |\mathcal{P}^-(p)| \to |\mathcal{P}^-(u)|$ for all $u \in B_\delta(p)$, where $|\mathcal{P}^-(p)|$ denotes the support of $\mathcal{P}^-(p)$. We wish now to show that $|\mathcal{P}^-(p)|^\circ = K^\circ \cap \Phi(p)^{-1}((-\infty, 0))$. Due to the continuity of $\Phi(u)$ it holds

that $K^\circ \cap \Phi(p)^{-1}((-\infty, 0)) \subseteq |\mathcal{P}^-(p)|^\circ$. Let now $x \in |\mathcal{P}^-(p)| \setminus (K^\circ \cap \Phi(p)^{-1}((-\infty, 0)))$, i.e., $\Phi(p)(x) = 0$. Since $\mathcal{P}(u)$ is a pure polyhedral complex, there is a $R \in \mathcal{P}_d(u)$ such that $x \in R$. It follows that $x \in H_{1,L+1,R}(p) \cap R$ with $\dim(H_{1,L+1,R}(p)) = d - 1$. If $\dim(H^1_{1,L+1,R}(p) \cap R) < d$, then there is a face $F \preceq R$ such that $F \subseteq H_{1,L+1,R}(p)$, which is a contradiction to $H^1_{1,L+1,R}(p) \cap R_0 = \emptyset$ and hence it holds that $\dim(H^1_{1,L+1,R}(p) \cap R) = d$. The latter fact implies that $\Phi(u)$ takes on exclusively positive values on $(H^1_{1,L+1,R}(p) \cap R)^\circ \neq \emptyset$ and hence for every open subset $U \subseteq \mathbb{R}^d$ with $x \in U$, it holds that $U \cap \Phi(p)^{-1}((0, \infty)) \neq \emptyset$. Thus, $x \notin |\mathcal{P}^-(p)|^\circ$ and hence $|\mathcal{P}^-(p)|^\circ = K^\circ \cap \Phi(p)^{-1}((-\infty, 0))$. Since $\mathcal{P}^-(p)$ and $\mathcal{P}^-(u)$ are isomorphic and $\Phi(u)$ is also topological stable due to Lemma 14 and Lemma 13, the same arguments can be applied in order to show $|\mathcal{P}^-(u)|^\circ = K^\circ \cap \Phi(u)^{-1}((-\infty, 0))$. Hence, since the restriction of $\varphi$ to the interiors $\varphi_{|\mathcal{P}^-(p)|^\circ} : |\mathcal{P}^-(p)|^\circ \to |\mathcal{P}^-(u)|^\circ$ is a homeomorphism as well, we conclude that $K^\circ \cap \Phi(u)^{-1}((-\infty, 0))$ and $K^\circ \cap \Phi(p)^{-1}((-\infty, 0))$ are homeomorphic. Let $F$ be any face of $K$ with $\dim(F) \neq 0$, then by the same arguments it follows that $F^\circ \cap \Phi(u)^{-1}((-\infty, 0))$ and $F^\circ \cap \Phi(p)^{-1}((-\infty, 0))$. Furthermore, due to the fact that $\Phi(p)$ is topologically stable and the choice of $u$, if $\dim(F) = 0$, it holds that $F \subseteq K \cap \Phi(p)^{-1}((-\infty, 0))$ implies that $F \subseteq K \cap \Phi(u)^{-1}((-\infty, 0))$ and hence

$$
\partial K \cap \Phi(p)^{-1}((-\infty, 0)) = \left( \bigsqcup_{\substack{F \preceq K, F \neq K \\ \dim(F) \neq 0}} F^\circ \sqcup \bigsqcup_{F \in K_0} F \right) \cap \Phi(p)^{-1}((-\infty, 0))
$$

$$
\cong \left( \bigsqcup_{\substack{F \preceq K, F \neq K \\ \dim(F) \neq 0}} F^\circ \sqcup \bigsqcup_{F \in K_0} F \right) \cap \Phi(u)^{-1}((-\infty, 0))
$$

$$
= \partial K \cap \Phi(u)^{-1}((-\infty, 0))
$$

Alltogether, we conclude that $K \cap \Phi(p)^{-1}((-\infty, 0))$ is homeomorphic to $K \cap \Phi(u)^{-1}((-\infty, 0))$. $\square$

We can finally show the stability of the constructed neural network for the lower bound of the topological expressive power.

**Proposition 10.** *There is an open set $U \subseteq \mathbb{R}^D$ in the parameter space of the architecture $(d, m \cdot d, \ldots, m \cdot d, w, 1)$ such that $\Phi(u)$ restricted to the unit cube has at least the same topological expressivity as $F$ in Theorem 24 for all $u \in U$.*

*Proof.* In order to obtain stability we first modify our construction in the following two ways:

- To ensure that all the resulting annuli in $F^{-1}((-\infty, 0))$ of different dimensions have positive distance to each other, we adjust $f^{(w_1, \ldots, w_{d-1})}$ by rescaling the summands of $f^{(w_1, \ldots, w_{d-1})}$, i.e., we make $F = \Phi(u)$ combinatorially stable with respect to the unit cube (c.f. Definition 13).

- Depending on the parity of the $w_k$ the "outermost annuli in $F^{-1}((-\infty, 0))$ around a cutting point" might be surrounded by a fulldimensional $0-$region. In order to guarantee stability, we transform the fulldimensional $0-$regions into slightly positive, but still constant regions by adding a small constant $b$, i.e., we make $F = \Phi(u)$ topologically stable with respect to the unit cube (c.f. Definition 14).

We achieve the first property by setting

$$
\hat{g}_q^{(k, w_{k-1})}(x) = \begin{cases} \max\{0, \mathbf{1}^T x\} & q = 0 \\ \max\{0, \mathbf{1}^T x - (\frac{1}{4}-)\} & q = w + 1 \\ \max\{0, 2(\mathbf{1}^T x - (2q - 1)/8w)\} & \text{else} \end{cases}
$$

for all $k = 2, \ldots, d$ and constructing $F$ dependent on $h^{(L, \mathbf{m}, d)}$ and the new $\hat{g}_q^{(k, w_{k-1})}(x), k = 2, \ldots, d$ in the same way as before. It is straighforward, that the the scaling does not change the

topology of the sublevel set, since it merely makes the annuli thinner. The second property we achieve by simply setting $F'(x) = F(x) + b$, where $b = \min_{k=2,\ldots,d}\{\frac{1}{8w_k \cdot M}\}$. We now argue, that $F^{-1}((-\infty, 0))$ and $F'^{-1}((-\infty, 0))$ are homeomorphic, since adding $b$ also just makes the annuli in $F^{-1}((-\infty, 0))$ thinner. Let $k \in [d]$ and $A$ be an $k-$annuli in $F^{-1}((-\infty, 0))$ and $A_k = p_k(A)$ be its projection onto the first $k$ coordinates. It follows that there is a cutting point $c \in \mathbb{R}^k$ such that $A_k = B^k_{q/(2w_k \cdot M)}(c) \setminus \overline{B^k_{(q-1)/(2w_k \cdot M)}(c)}$ for a suitable $q = 1, \ldots, w_k$. Since $b \leq \{\frac{1}{8w_k \cdot M}\}$ it follows that

$$A'_{k,c,q} := B^k_{(q-2b)/(2w_k \cdot M)}(c) \setminus \overline{B^k_{(q-1+2b)/(2w_k \cdot M)}(c)}$$

is also a $k$-annuli and revisiting the proof of Lemma 19 reveals that $F(A'_{k,c,q} \times \mathbb{R}^{d-k} \cap K) = (-\frac{1}{4w_k}, -b)$ and hence $F'(A'_{k,c,q} \times \mathbb{R}^{d-k} \cap K) = (-\frac{1}{4w_k} + b, 0)$ since $\frac{1}{8w_k} \geq b$. We conclude that for every $k$-annuli in $F^{-1}((-\infty, 0))$ there is an $k$-annuli in $F'^{-1}((-\infty, 0))$ and since it clearly holds that $\mathrm{sgn}\, F(x) = \mathrm{sgn}\, F'(x)$ for all $x \in F^{-1}((0, \infty))$, it follows that $F^{-1}((-\infty, 0))$ and $F'^{-1}((-\infty, 0))$ are homeomorphic. Let $p \in \mathbb{R}^D$ such that $\Phi(p) = F'$. Then, since $\Phi(p)$ is topologically stable it follows by Proposition 26 that there is an open set in $\mathbb{R}^D$ containing $u$ such that $\Phi(u)^{-1}((-\infty, 0)) \cap K$ is homeomorphic to $\Phi(p)^{-1}((-\infty, 0)) \cap K$ for all $u \in U$, where $K$ is the unit cube.

$\square$

## 5.4 Upper bound

In this section we will provide a formal proof for the upper bounds.

**Proposition 3.** *Let $F \colon \mathbb{R}^d \to \mathbb{R}$ be a neural network of architecture $(d, n_1, \ldots, n_L, 1)$. Then it holds that $\beta_0(F) \leq \sum_{(j_1,\ldots,j_L) \in J} \prod_{\ell=1}^L \binom{n_\ell}{j_\ell}$ and for all $k \in [d-1]$ that*

$$\beta_k(F) \leq \binom{\sum_{(j_1,\ldots,j_L) \in J} \prod_{\ell=1}^L \binom{n_\ell}{j_\ell}}{d-k},$$

*where $J = \big\{(j_1, \ldots, j_L) \in \mathbb{Z}^L \colon 0 \leq j_\ell \leq \min\{d, n_1 - j_1, \ldots, n_{\ell-1} - j_{\ell-1}\}$ for all $\ell = 1, \ldots, L\big\}$.*

*Proof.* Theorem 1 in (Serra et al., 2017) states that $F$ has at most $\sum_{(j_1,\ldots,j_L) \in J} \prod_{l=1}^L \binom{n_l}{j_l}$ linear regions. Let $\mathcal{P}$ be the canonical polyhedral complex of $F$ and let $\mathcal{P}_{k+1} = \{P \in \mathcal{P} \mid \dim(P) \leq k+1\}$ be the $k+1$-skeleton of $\mathcal{P}$. Furthermore let $\mathcal{P}^-_{k+1} := \{P \cap F^{-1}((-\infty, 0]) \mid P \in \mathcal{P}_{k+1}\}, \mathcal{P}^+_{k+1} := \{P \cap F^{-1}([0, \infty)) \mid P \in \mathcal{P}_{k+1}\}$ be the subcomplexes of the subdivision of $\mathcal{P}_{k+1}$ where $F$ takes on exclusively non-positive respectively non-negative function values and $\mathcal{P}^=_{k+1} := \{P \cap F^{-1}(0) \mid P \in \mathcal{P}_{k+1}\}$. Let $\mathcal{P}^0_{k+1} := \big(\mathcal{P}^=_{k+1} \cap \mathcal{P}^-_{k+1}\big) \setminus \mathcal{P}^+_{k+1}$ be the set of polyhedra where $F$ takes on function value 0 and they are not exclusively bounded by elements of $\mathcal{P}^-_k$. We can lift every element in $\mathcal{P}^0_{k+1} \subseteq \mathcal{P}^-_{k+1}$ onto a higher dimension such that

(i) the poset structure of the lifted polyhedral complex agrees with the initial poset structure restricted to $\mathcal{P}^0_{k+1}$,

(ii) the maximal (w.r.t. inclusion) polyhedra are lifted into polyhedra of dimension $k+1$, and

(iii) if two polyhedra are included in the same polyhedron, their dimensions are increased by the same amount, that is, the codimension of the maximal polyhedron $P \in \mathcal{P}^0_{k+1}$ they are both included in.

We denote by $\tilde{\mathcal{P}}^0_{k+1}$ the polyhedral complex of the thickened up polyhedra and by $\mathcal{C}$ the polyhedral complex that results my modifying from $\mathcal{P}^0_{k+1}$ in the polyhedral complex $\mathcal{P}^-_{k+1} \cup \mathcal{P}^+_{k+1}$ in the same way and adjusting the position of the remaining polyhedra. For every element $P \in \mathcal{P}_{k+1}(k+1)$ at most one of the two following conditions can hold due to the fact that $F$ is affine linear on $P$:

(i) $P \cap F^{-1}((-\infty, 0])$ is adjacent to a maximal polyhedron in $\tilde{\mathcal{P}}^0_{k+1}$, or

(ii) $P \cap F^{-1}([0, \infty))$ is a $(k+1)$-face of $\mathcal{P}_{k+1}^+$.

Let $\mathcal{D} := \tilde{\mathcal{P}}_{k+1}^0(k+1) \cup \mathcal{P}_{k+1}^+(k+1)$, then it follows that

$$\#\mathcal{D} \leq \#\mathcal{P}_{k+1}(k+1) \leq \binom{\sum_{(j_1,\ldots,j_L)\in J} \prod_{l=1}^L \binom{n_l}{j_l}}{d-k}.$$

The second inequality follows from considering the $(k+1)$-faces as the intersection of $d-k$ linear regions (which are of dimension $d$). Furthermore, by construction it holds that $\beta_k(F) = \beta_k(|\mathcal{C} \setminus \mathcal{D}|)$, where $|\mathcal{C} \setminus \mathcal{D}|$ denotes the support. To see this, note that $F^{-1}((-\infty, 0)) = \bigcup\{P^\circ \mid P \in P_d^- \setminus P_d^=\} \cup \bigcup\{P_d^-(0) \setminus P_d^=(0)\}$ and $F^{-1}((-\infty, 0)) \cap \mathcal{P}_k^- = \bigcup\{P^\circ \mid P \in \mathcal{P}_k^- \setminus \mathcal{P}_k^=\} \cup \{\mathcal{P}_k^-(0) \setminus \mathcal{P}_k^=(0)\}$, which implies that $\beta_k(F) = \beta_k(F^{-1}((-\infty, 0)) \cap \mathcal{P}_k^-)$. The construction of $\mathcal{C}$ and $\mathcal{D}$ ensure that $F^{-1}((-\infty, 0)) \cap \mathcal{P}_k^-$ is homotopy equivalent to $|\mathcal{C} \setminus \mathcal{D}|$.

For the sake of simplicity, we compute $\beta_k(|\mathcal{C} \setminus \mathcal{D}|)$ using cellular homology. Ideally, we would like to equip $\mathcal{C}$ with a canonical CW-complex structure, i.e., the $k$-cells of the CW-complexes precisely correspond to the $k$-faces of the respective polyhedral complex, and attachment maps are given by face incidences. However, $\mathcal{C}$ contains unbounded polyhedra. In particular, an unbounded polyhedron cannot correspond to a CW-cell. Conveniently, one can sidestep this issue by observing the following: There exists a large enough number $M$ such that $\{P \cap [-M, M]^n : P \in \mathcal{C}\}$ is homotopy equivalent to $\mathcal{C}$, (by deformation retracting the unbounded faces to their restrictions under $[-M, M]^n$). In particular, we may assume without loss of generality that $\mathcal{C}$ is a compact polyhedral complex equipped with a finite CW-complex structure as described above.

We show that $\beta_k(|\mathcal{C} \setminus \mathcal{D}|) \leq \#\mathcal{D}$ by induction on the number of $(k+1)$-faces of $\mathcal{D}$. If $\mathcal{D}$ has no $(k+1)$ faces, then the inequality holds vacuously, since $\beta_k(|\mathcal{C}|) = 0$.

When proving the induction step, we proceed to delete full-dimensional polyhedra, resulting in the creation of new polyhedral/CW-complexes. For our purposes, deleting the polyhedra accounts to deleting the smaller-dimensional faces as well that bound no other full-dimensional polyhedra (which we call *redundant* in the following), we delete such faces as well.

To show the induction step, let $\mathcal{D} = \mathcal{D}' \cup \{P\}$. By the induction hypothesis we know that $\beta(|\mathcal{C} \setminus \mathcal{D}'|) \leq \#\mathcal{D}'$. Let $B$ be the support of $\mathcal{C} \setminus \mathcal{D}$ and $A$ the support of $\mathcal{C} \setminus \mathcal{D}'$.

Our goal is to embed the cellular homology group $H_k(B)$ into $\mathbb{Z} \oplus H_k(A)$. Such an embedding readily implies that $\beta_k(B) \leq 1 + \beta_k(A)$. From this, the induction step follows:
$$\beta_k(B) \leq 1 + \beta_k(A) \leq 1$$

To prove the induction step, we first delete the $k+1$-dimensional face itself (that is, without deleting the redundant faces, resulting in a polyhedral complex whose support we denote by $B'$), and observe by the elaborate definition of cellular homology groups that this induces a map $\phi_1 \colon H_k(B') \to \mathbb{Z} \oplus H_k(A)$. One can additionally observe that this map is an embedding: Notice that the homology arises from the (relative) homologies of the chain complex

$$\ldots \to C_{k+1} \xrightarrow{\partial_{k+1}} C_k \xrightarrow{\partial_k} C_{k-1} \to \ldots$$

Deleting the $k+1$-face decreases the image of the boundary map $\partial_{k+1}$ and hence increases the $k$-th homology; however, it is straightforward to observe that

$$\phi_1 \colon H_k(B') \cong \ker \partial_k / \operatorname{im} \partial_{k+1} \to [\sigma] \oplus \ker \partial_k / \operatorname{im} \partial_{k+1} \cup [\sigma] \cong \mathbb{Z} \oplus H_k(A)$$

which maps a homology class $[\sum_{\tau \in C_k} c_\tau \tau]$ from the domain to $(c_\sigma, [\sum_{\tau \in C_{k/[\sigma]}} c_\tau \tau])$ is an embedding, where $\sigma$ is the generator corresponding to the deleted $k+1$-face.

To finish off the construction of the embedding, we finally define $\phi_2 \colon H_k(B) \to H_k(B')$, i.e., the map induced by deleting maximal faces of $B'$ formerly incident to the deleted polyhedron. This operation might reduce the kernel of the $k$-th boundary map, and hence potentially decrease the $k$-th Betti number. It is, however, again straightforward to observe that the map is injective in any case, in a similar fashion as above.

The claimed embedding is now $\phi_1 \circ \phi_2$, finishing the proof for $k \in [d-1]$.

$\square$

