# OpenReview forum: "Topological Expressive Power of ReLU Neural Networks"
_ICLR.cc/2024/Conference — Submitted to ICLR 2024_

### Official Review · Reviewer_XYHA · 2023-10-31

**Soundness:** 2 fair
**Presentation:** 2 fair
**Contribution:** 2 fair
**Rating:** 3
**Confidence:** 4

**Summary:**

This article is about topology of certain sublevel sets of functions defined by fully connected ReLU neural networks.  Asymptotic bounds for Betti numbers of the sublevel sets of such functions are established.

**Strengths:**

Paper is situated within the framework  trying to approach expressivity of neural networks via topology. A set of interesting mathematical results concerning bounds for Betti numbers is proven

**Weaknesses:**

Weaknesses:

1)In deep learning  both the positive and negative data points typically lie near very low dimensional surfaces, so in general, there is no relation between Betti numbers of the sublevel set of the function defining a decision boundary and the Betti numbers of the support of the distributions from which the data points are sampled. The Betti numbers of the support of distribution can be arbitrary big, while at the same time the zeroth Betti number of the sublevel set of the function defining a decision boundary can be  equal to one. Therefore the practical meaning of the paper's results is limited.


2)Lower bound is obtained only for some specific weights of neural network. However, given a neural network architecture there are always weights that produce constant function and thus have strictly zero Betti numbers, so the meaning of  paper's "lower bound" term is not quite clear.

3)Also it is not clear whether the constructed network weights can be  found via regular optimization algorithms,

4)The calculation of Betti numbers is difficult so it  also undermines practical implications of the work.

5)The claim of the exponential gap is somewhat unclear in the paper.

6)The upper bound proof lacks some details, only about half a page is devoted to upper bounds, the paper mostly concerned with lower bounds.

7)The lower bounds found by the authors are similar to previous ones that have appeared in the literature, e.g. in  Bianchini and Scarselli (2014). For example the principal zeroth Betti number result is an extension of the loc.cit to  ReLU activations.

Minor remarks:

Grammar errors: section 1.2.1 : ares -> are

Notations are somewhat confusing M_a and M_b are topological spaces, but M is an integer in section 2, lemma 4  etc.

Why does this graph, consisting of two points, represents the functions that folds the interval on Figure 1, there seems to be a problem with this figure, are some lines missing ?

**Questions:**

Can authors provide an example from real world data when their bounds have practical implications ?

The paper really needs to show concrete real world examples  with practical meaning of the paper's results.

Reply to post-rebuttal remarks:
I thank the authors for further clarification. I encourage the authors to spell out the precise meaning of their notion "lower bound of" in concrete mathematical terms. It seems that this is the lower bound of the maximum of certain quantity taken over some set, but what exactly is the set over which the maximum is taken? Does this maximum depends on the architecture only, or also on the dataset $X$? If it depends on the dataset $X$ what is the meaning/effectiveness of this depending on $X$ notion?
Also concerning another issue, I really encourage the authors to try to find a practical real-world case in which their results bring a novel perspective.

---

> ### Author Response · Authors · 2023-11-17
>
> We thank the reviewer for their valuable input. We would like to address their concerns in the same order:
>
> 1. In such a case, if $X = X^- \cup X^+$ denotes the union of the supports of the data distributions of the positive and the negative data points, the Betti numbers of $X \cap F^{-1}((-\infty,0])$ need to equal the Betti numbers of $X^-$. The construction of the lower bound is easily applicable in such a setting as well.
> 2. We are motivated to define the topological expressive power of an architecture as the representation of the "most complex" topology achievable by a neural network with that architecture. Consequently, our construction provides a lower bound for this concept.
> 3. Proposition 10 suggests that in the parameter space, there are open neighborhoods around the weight vectors that arise from our construction such that every point in these open neighborhoods also represent functions with Betti numbers at least our bounds. Therefore, it is possible to find the weight representation of such a function in the parameter space by e.g. gradient descent methods.
> 4. The main goal of this paper is not to suggest techniques relevant for practice, but rather to point out at theoretical barriers (or capabilities) of expressivity of a neural network. Furthermore, the topological theoretical insights might have connection to other aspects of deep learning, see e.g. the discussion with reviewer UFRF.  Moreover, Betti numbers are indeed used in practice, at least in dimensions up to 2.
> 5. The ''exponential gap'' refers to the fact that only exponentially sized shallow neural networks can encapsulate the Betti numbers that are already achievable by linear sized deep neural networks.  We can gladly elaborate further if the reviewer wishes to.
> 6. We have tried to keep the lower bound construction self-containing, which required multiple pages since definitions and figures had to be introduced. On the other hand, we have included a sketch for our upper bounds, missing steps being standard topological arguments. These can be found in the appendix.
> 7. While the extension from sigmoid activation to ReLU activation may seem incremental, it is still a significant contribution since the ReLU activation function is widely used in modern neural networks. Neither in Bianchini-Scarselli nor in our case is it immediately clear that one must imply the other.  Moreover, we have extended the result to any Betti number, answering the question whether the expressivity results hold as well for ``holes'' affirmatively for any dimension.
>
> We thank the reviewer once again, for the minor remarks in particular. Among others, the figure seems to have suffered a compilation error, which has been cleared, so that it shows the piecewise linear function $f(x)=0.5-|x-0.5|$ which we used in the first $L-1$ layers repeatedly to create the exponential number of ''holes''. We are looking forward to a discussion.

---

> > ### Comment · Reviewer_XYHA · 2023-11-23
> >
> > I thank the authors for the response. Some of concerns are addressed however several principal issues concerning in particular the practical applications of the proposed method remain.  Also, I do not quite understand the response to the 1st point, the intersection seem to coincide  with $X^{-}$ itself, on the other hand the paper results are not directly applicable to such intersection and it is not clear how they can be extended here.

---

> > > ### Author Response · Authors · 2023-11-23
> > >
> > > We thank the reviewer for the response and the comments.
> > > The intersection $X \cap F^{-1}((-\infty,0])$ does not coincide with $X^-$. The former represents the subset of $X$ mapped to the non-negative real line by $F$, while the latter is the set of negatively labeled data points. The construction of the lower bound is applicable in such a setting as well: If $k$ is the dimension of the manifold $X$, we have a $k$-dimensional cube in $X$, where our construction can be applied in the same manner. We acknowledge that the upper bound is indeed not directly applicable and depends more on the properties of the manifold $X$. Investigating upper bounds depending on $X$ would be an interesting follow-up research direction for our results. Furthermore, we want to emphasize that we do not propose a method for immediate practical applications. Instead, our focus is on exploring the fundamental nature of ReLU neural networks. We believe that our results, in their generality, contribute to theoretical applications, such as offering a stronger depth-distinguishing inapproximability result, as discussed with reviewer UFRF.

---

> ### Comment · Reviewer_XYHA · 2023-11-23
> **reply to post-rebuttal message**
>
> I thank the authors for further clarification. I  encourage the authors to spell out the precise meaning of their notion "lower bound of"  in concrete mathematical terms. It seems that this is the lower bound of a maximum of certain quantity taken over some set, but what exactly  is the set over which the maximum is taken? Does this maximum depends on the architecture only, or also on the dataset $X$?  If it depends on the dataset $X$ what is the meaning/effectiveness of this notion depending on $X$?
>
> Also concerning another issue, I really encourage the authors to try to find a practical real-world case in which their results bring a novel perspective.

---

### Official Review · Reviewer_UFRF · 2023-11-01

**Soundness:** 3 good
**Presentation:** 3 good
**Contribution:** 3 good
**Rating:** 8
**Confidence:** 4

**Summary:**

The paper studies ReLU networks from the perspective of their topological expressivity. The measure used here is that of Betti numbers that is a suitable measure for characterizing how complicated the topological properties of a network are.

The main contribution of the paper is to derive several upper and lower bounds for the Betti numbers depending on the depth and width of the ReLU network, by using clever constructions of functions.

The main takeaway is that Betti numbers depend on the depth, and can significantly grow with the depth. If the depth is unbounded, Betti numbers increase exponentially with the size of the network. In contrast, if the networks is shallow then its Betti numbers do not grow as fast. This is interesting as it showcases that a possible bottleneck for effective data representation is the depth.

The constructions in the paper are heavily inspired by previous ideas used in Montufar et al. where the goal was to characterize another measure of ReLU neural network complexity, that of linear regions. We know that the number of linear regions can exponentially grow with the depth, but not the width. The paper under review essentially sets out to formally establish the connection and proposes clever constructions to transfer the results to the complexity measure of Betti numbers.

**Strengths:**

+well-motivated theoretical question about the expressivity

+the question has been empirically observed and the paper develops interesting theory to address this in a simple binary classification setting

+in my opinion, the paper proves a very elegant characterization for expressivity and interesting dependence on Betti numbers for the depth and width

+potential interesting connections to dynamical systems (see comments below)

**Weaknesses:**

Overall, the paper is strong and there are not major weaknesses in my opinion. One thing I believe should pointed out though has to do with the novelty of the final conclusion of the paper.

- The key takeaway of the paper is that depth is more important than width. The paper has an elegant way of proving this via the Betti numbers. However, the reviewer just wants to point out that similar depth-width tradeoffs were known, albeit using different techniques and different connections. So in some sense we already knew that depth is exponentially better than width. For example:

The authors cite Telgarsky's works who used a basic triangle construction and as a measure of complexity he used the number of linear regions. Similarly, Montufar et al. had the number of linear regions as a way to show that depth is much more important.

There is also a generalization of the works of Telgarsky that use connections to dynamical systems (Li-Yorke chaos, periodic orbits) and the notion of *topological* entropy [3]. See [1], [2], [3]. Papers [1] and [2] give lower bound constructions using more general functions that than Telgarsky's triangle and [3] provides characterization using topological entropy.

[1] Depth-WidthTrade-offs for ReLU Networks via Sharkovsky’s Theorem
[2] Better depth-width trade-offs for neural networks through the lens of dynamical systems
[3] Expressivity of Neural Networks via Chaotic Itineraries beyond Sharkovsky’sTheorem

It would be interesting to see if the characterization of the Betti numbers for the depth/width tradeoffs can actually follow in certain cases because of the connection to Li-Yorke chaos and periodic points.

**Questions:**

Q: Related to the weakness comments above, do the authors see any connection between their construction and the notion of periodic points/topological entropy in dynamical systemts? At least their examples in Fig. 3,4,5,6 for the binary classification problem resembles both Telgarsky's triangle characterization, and also the more general result proved in [1] Depth-WidthTrade-offs for ReLU Networks via Sharkovsky’s Theorem.

Q: For 1-dimensional neural networks (i.e. input is just a real number) similar to the ones that Telgarsky used, do your results imply the exact separation that Telgarsky proved? Is there a sense why your results are stronger in this special case? I believe this is the simplest case where we can understand whether or not the connection to dynamical systems is valid.

**Details Of Ethics Concerns:**

-

---

> ### Author Response · Authors · 2023-11-17
>
> We would like to thank the reviewer for the insightful review and especially for pointing at potential connections between our construction and concepts in dynamical systems used in the referenced works, as well as to the construction in Telgarksys work.
>
> Regarding the first question, there are connections between our construction and the ones used in the mentioned references. For example, the coordinates of the cutting points (the points in our constructions that are the center of the resulting annuli) are non-periodic points of the simple triangle function(they are the points that get mapped to the fixed point $0$ after a certain number of application). Although this observation suggests a potential link between our construction and concepts in dynamical systems, we are not seeing a connection to the notion of topological entropy so far.
>
>
> Regarding the second question, for $d=1$, our bounds imply the depth separation given by Telgarsky and in fact even for shallower networks (i.e., the functions inapproximable by networks of depth $k$ and polynomial size(in $k$) can be represented by networks with a depth and size of $O(k^2)$). Since we showed that our construction is robust with respect to small perturbation, this separation holds for a full Lebesgue measure set of neural networks.
> We believe that the reason why the bounds obtained in this way are slightly better is due to the simplicity of the analysis. Since all our connected components have a constant size and contain a point with a constant negative function value, the inapproximability by small shallow neural networks follows easily by the fact that they are not capable of expressing the same number of connected components.
> For general $d$, our construction differs from the one of Telgarsky in the way that we apply the triangle map in every coordinate and make use of the arising subdivision of the unit cube.
>
>
>
> Due to the technical overlaps, we totally agree with the reviewer that it would be interesting to see whether we can use a possible connection to concepts from dynamical systems like Li-Yorke chaos and periodic points to gain a better understanding of the topological behaviour of the decision regions. On the other side, topological results like ours might also be fruitful for dynamical systems or further inapproximability results and we believe that exploring this connection more explicitly and rigorously is an interesting direction for future research.
>
> We would like to thank the reviewer once again for the meaningful and interesting review and we would be happy to answer further questions or discuss potential deeper connections to dynamical systems.

---

### Official Review · Reviewer_o7g5 · 2023-11-02

**Soundness:** 2 fair
**Presentation:** 2 fair
**Contribution:** 2 fair
**Rating:** 6
**Confidence:** 1

**Summary:**

The paper studies the expressivity of ReLU neural networks in the setting of a binary classification from a topological perspective.
The authors prove new lower and upper bounds for topological expressivity of ReLU networks. Here, the topological expressivity is the sum of Betti numbers of input subspaces, which network separates. Such expressivity grows polynomially with the width (for fixed depth) and exponentially with the depth (for fixed width). Most of the paper is dedicated to obtaining the lower bound by explicitly constructing weights of a network.

**Strengths:**

Research on the intersection of topology and deep learning is active right now.
Regarding the expressivity analysis and proving UAT-like theorems, I am not an expert in this area and I can't evaluate originality and impact of the manuscript.
I haven't thoroughly checked math, but I don't see evident errors.
Overall, the paper is well written and language is fine.

**Weaknesses:**

1. I don't understand the notation $\beta_0(F) \in \Omega(M^d \cdot n_L)$. Is it the same as $\beta_0(F) = \Omega(M^d \cdot n_L)$ ? (that is, $C_1 M^d \cdot n_L \le \beta_0(F) \le C_2 M^d \cdot n_L$
2. The most of the paper is dedicated to the proof of the **existence** of a network with a given topological expressivity.
But in deep learning we are interested in a practical algorithm for finding such a network.
The manuscript will benefit from computational experiments. You can use simple synthetic datasets with known Betti numbers (like in Naitzat et al. (2020)) and estimate depth/width of a network which is able to classify it with accuracy > 0.95, for example.
3. The manuscript is very long (30 pages), the Appendix is dedicated to proofs. Maybe some journal will be a better destination for such a manuscript.

**Questions:**

1. You explicitly construct a network with a given topological expressivity, but are this network's weights reachable by gradient optimization?

---

> ### Author Response · Authors · 2023-11-17
>
> We thank the reviewer for their suggestions, and would like to address their concerns in the same order.
>
> 1.  One standard definition of Landau symbols which is used often in complexity theory is that, given a function $f$, the terms $O(f)$ resp. $\Omega(f)$ denote sets for which $g=O(f)$ resp. $g=\Omega(f)$ (as the reviewer mentioned) would hold, see e.g. https://en.wikipedia.org/wiki/Big_O_notation#Matters_of_notation. We can, however, gladly switch to the equal sign, as it seems to be the better-known variant.
> 2. Proposition 10 of the paper implies that there exist positive radius neighborhoods around each of our constructed functions that satisfy the same properties with respect to Betti numbers of the sublevel sets. Therefore, efficient algorithms based on e.g. gradient descent methods can be used for the detection of functions that prove our lower bounds (which also answers the reviewer's question). That being said, we agree that experiments would be an enrichment to our theoretical findings.
>
> 3. We believe that the short version of the paper is self-contained (at least with respect to the definitions and constructions) and includes all our results, and is therefore suitable for a conference. The interested reader shall be able to find the appendix online in the future.
>
> We thank the reviewer once again and are looking forward to a discussion.

---

### Meta-Review · Area_Chair_46Aj · 2023-12-05

**Metareview:**

The paper studies the *topological expressivity* of a ReLU network $F$ by means of the Betti numbers of the sublevel set $F^{-1}(-\infty,0)$. The main result concerns an (asymptotic) lower bound of the individual maximal Betti numbers attainable for a given number of layer and widths (no restriction is made on connectivity between adjacent layers). Together with an upper bound derived using results on linear regions (Serra et al., 2017), the authors establish an exponential gap between shallow and deep neural networks measured by the said notion of topological expressivity.

The reviewers found the theoretical question well-motivated, and the main theoretical results addressing a more refined notion of topological expressivity, namely, individual Betti numbers rather than their sum (Bianchini and Scarselli (2014)), interesting and elegant. However, some of the reviewers maintained that empirical evidence supporting the theoretical findings (e.g., topological expressivity of networks trained on real-world datasets) is needed for the paper to be recommended for acceptance; the issue remained despite the discussions with the authors. The authors are highly encouraged to incorporate the important feedback given by the knowledgeable reviewers, and resubmit.

**Justification For Why Not Higher Score:**

Writing problems were raised, but mainly due to the reviewers' claims concerning lack of numerical work.

**Justification For Why Not Lower Score:**

N\A

---

### Decision · Program_Chairs · 2024-01-16

Reject